

# Hilbert space fragmentation in a 2D quantum spin system with subsystem symmetries

Alexey Khudorozhkov[1,2], Apoorv Tiwari[3,4], Claudio Chamon[5] and Titus Neupert[3]

**1** Physics Department, Boston University, Boston, MA 02215, USA
**2** Department of Physics, ETH Zurich, 8093 Zurich, Switzerland
**3** Department of Physics, University of Zurich, Winterthurerstrasse 190,
8057 Zurich, Switzerland
**4** Condensed Matter Theory Group, Paul Scherrer Institute,
CH-5232 Villigen PSI, Switzerland
**4** Physics Department, Boston University, Boston, MA 02215, USA

## Abstract

We consider a 2D quantum spin model with ring-exchange interaction that has subsystem symmetries associated to conserved magnetization along rows and columns of a square lattice, which implies the conservation of the global dipole moment. The model is not integrable, but violates the eigenstate thermalization hypothesis through an extensive Hilbert space fragmentation, including an exponential number of inert subsectors with trivial dynamics, arising from kinetic constraints. While subsystem symmetries are quite restrictive for the dynamics, we show that they alone cannot account for such a number of inert states, even with infinite-range interactions. We present a procedure for constructing shielding structures that can separate and disentangle dynamically active regions from each other. Notably, subsystem symmetries allow the thickness of the shields to be dependent only on the interaction range rather than on the size of the active regions, unlike in the case of generic dipole-conserving systems.



# 1   Introduction

A question that provokes a lot of research in the field of non-equilibrium dynamics of quantum systems is whether an isolated quantum system eventually comes to an equilibrium ("thermalizes"). An isolated quantum system is said to thermalize if in the thermodynamic limit and at long times, for any small subsystem the rest of the system acts as a thermal bath [1]. The renowned Eigenstate Thermalization Hypothesis (ETH) [2–4] implies that in generic thermalizing quantum systems all eigenstates in the bulk of the spectrum are thermal, meaning that average values of local observables saturate to values dictated by a thermal ensemble at the temperature set by the eigenenergy [5].

Two widely known classes of non-thermalizing systems, where ETH does not hold, are integrable systems and many-body localized (MBL) systems. In both cases ergodicity is prevented by the existence of an extensive number of conserved quantities. In the former case, they are directly encoded in the Hamiltonian, while in the latter case, emergent local integrals of motion are created by strong disorder. Apart from these two instances, ETH was believed to hold true in generic non-integrable models and has been extensively tested numerically [4, 6–14]. However, more recently a possibility for eigenstates to behave non-thermally in non-integrable disorder-free many-body quantum systems has been observed experimentally [15] and explored theoretically, both in 1D [16–40], 2D [21, 31, 41–44] and higher dimensions [21, 45–48]. These states have been dubbed "quantum many-body scars".

One of the possible mechanisms leading to quantum scars is the *Hilbert space fragmentation* [22–24, 29, 30, 33–35, 43, 49], when the Hilbert space is fractured into dynamically disconnected sectors (further we call them "fragments") due to kinetic constraints, without any obvious conserved quantities corresponding to these sectors. While initial works on quantum scars considered models where non-thermal eigenstates constituted a vanishing fraction of the spectrum, models with Hilbert space fragmentation showed the possibility of exponentially many disconnected fragments [23, 24]. In particular, in Ref. [23], where such extensive fragmentation was dubbed "Hilbert space shattering", the authors studied a 1D spin model with global charge and dipole conservation laws and on general grounds argued that dipole-conserving models in any dimension should exhibit Hilbert space shattering, including an exponential number of completely inert states. Our work presents an example of the Hilbert space shattering in 2D.

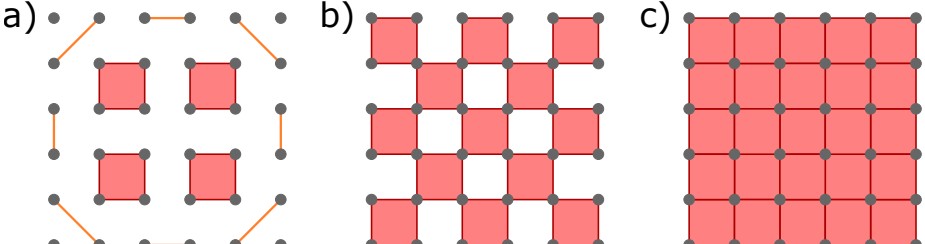

Figure 1: Three spin-1/2 models on the square lattice with ring-exchange interaction of the form $S^+S^-S^+S^-$ on the red solid plaquettes. The model (a) has been proposed in Ref. [57] and shown to exhibit the SSPT phase that is equivalent to a HOTI. The ring-exchange is present on every fourth plaquette, while at the edges the pairwise interaction is XY ($S^+S^-$), as shown with orange lines. The model (b), where the ring-exchange dynamics is allowed for every second plaquette, is an equivalent representation of the QLM (or, to be precise, its kinetic part), as well as the fully packed QDM if considered in a particular charge sector. The XY-plaquette model (c) is a natural continuation of this sequence, that enables ring-exchange dynamics on *every* plaquette of the square lattice.

In this paper, we numerically study the spin-1/2 XY-plaquette model: a 2D dipole-conserving model with additional subsystem symmetries along columns and rows of the square lattice. The model consists of the ring-exchange terms (1) on each elementary plaquette of the lattice. While the low-energy properties of similar models with ring-exchange terms have been studied in Refs. [50–55], the properties of the excited states, especially with regards to ergodicity, have not been investigated so far, to the best of our knowledge. The motivation behind studying ergodic properties of this model is twofold. For one, the XY-plaquette model conserves dipole moment and, as already mentioned, should exhibit an extensive Hilbert space fragmentation [23]. Secondly, recent works studied several related models with subsystem symmetries on the square lattice that have more restrictive dynamics than the XY-plaquette model. In particular, we compare in Fig. 1 a family of three models, which essentially differ in the density of the plaquettes to which the ring exchange term is applied. The model in Fig. 1(a), which is a 2D generalization of the 1D AKLT spin chain [56], has been shown to exhibit a subsystem symmetry protected topological phase (SSPT) with gapless corner modes, which is an instance of a higher-order topological insulator (HOTI) [57]. This model has the ring-exchange dynamics enabled on every fourth plaquette of the square lattice. Furthermore, as illustrated in Fig. 1(b), the kinetic part of the quantum link model (QLM) [58, 59] can be represented in a similar form, with the ring-exchange on every second plaquette (see Appendix D for details on how to rewrite QLM in this form). The fragmentation of the Hilbert space in the QLM has been recently observed in Ref. [44]. In addition, the fully packed quantum dimer model (QDM) [60] is equivalent to a particular charge sector of the QLM (as we show in Appendix E), and exhibits non-ergodic properties as have been shown in Refs. [61, 62]. Three of the above-mentioned models (HOTI phase from [57], QLM and QDM) not only have subsystem symmetries, but even more restrictive local symmetries (as explained in Appendices D–E). It is therefore the next logical step to lift these local conservation laws and allow the ring-exchange dynamics on *every* plaquette of the square lattice [Fig. 1(c)], such that the most restrictive integrals of motion are subsystem symmetries.

The structure of the paper is as follows. In Section 2, we introduce the model and its symmetries, and explain how it can be efficiently studied numerically. In Section 3, we explicitly demonstrate that the model exhibits Hilbert space fragmentation and discuss how this is altered by longer-range interactions. For different ranges of interaction, we count the number

of inert states using the transfer matrix approach and show that for the minimal interaction range it scales exponentially with the number of degrees of freedom (d.o.f.), unlike for the case of infinite interaction range, where the Hilbert space fragmentation is absent and the number of inert states scales sub-exponentially. We therefore conclude that subsystem symmetries alone are not enough for the shattering of the Hilbert space. Next, we numerically investigate the structure of the full fragmented Hilbert space for lattices up to 5×5. We show the size distribution of the fragments, as well as elaborate on which symmetry sectors are "the most fragmented". The abundance of low-dimensional fragments clearly demonstrates that the model violates ETH in the strong sense. In Section 4, we present the procedure for constructing shielding structures that lead to isolated disentangled active regions. Notably, subsystem symmetries allow the thickness of these shields to be comparable to the interaction range, unlike in the case of generic dipole-conserving systems where a shield has to be at least of the same size as the active region it isolates [23]. In Section 5, we study the level spacing statistics and the entanglement entropy of eigenstates. We show that while in the absence of disorder the model exhibits Poisson spacing statistics, upon the addition of "infinitesimal" disorder the statistics changes to Wigner-Dyson, while the fragmentation structure is preserved. We therefore conclude that the XY-plaquette model with "infinitesimal" disorder is a non-integrable model that nevertheless violates ETH. In addition, we study the time evolution of the entanglement entropy in different fragments and for different interaction ranges.

## 2 XY-plaquette model

We study a 2D quantum model with a spin-1/2 degree of freedom at each site of a square lattice. The Hamiltonian contains ring exchange terms defined on elementary square plaquettes.

$$\hat{H}_{1\times1} = K \sum_{\langle ijkl\rangle \in \square_{1\times1}} \left( \hat{S}_i^+ \hat{S}_j^- \hat{S}_k^+ \hat{S}_l^- + \text{h.c.} \right), \tag{1}$$

where $\hat{S}_i^{x,y,z}$ are spin-1/2 Pauli operators acting on the Hilbert space at site $i$, $\hat{S}_i^{\pm} \equiv \hat{S}_i^x \pm i\hat{S}_i^y$ and $\square_{1\times1}$ is the set of elementary plaquettes, i.e., squares of size $1\times1$ lattice spacings (as illustrated in Fig. 2, left panel). In addition to spatial ($C_4$ rotation, mirror reflection, translation) and time reversal symmetries, this model possesses $U(1)$ subsystem symmetries generated by the operators $\hat{U}_n^x, \hat{U}_m^y$ which act non-trivially on 1D subsystems, i.e., columns $\Lambda_n^x$ and rows $\Lambda_m^y$.

$$\hat{U}_n^x(\alpha) = \exp\left( i\alpha \sum_{j\in\Lambda_n^x} \hat{S}_j^z \right), \quad n = 1,\ldots,N_x,$$
$$\hat{U}_m^y(\alpha) = \exp\left( i\alpha \sum_{j\in\Lambda_m^y} \hat{S}_j^z \right), \quad m = 1,\ldots,N_y, \tag{2}$$

where $N_x$ and $N_y$ are the number of columns and rows of the square lattice respectively. These symmetries correspond to conserved magnetization on each column and row, which can be chosen as good quantum numbers (for convenience, we will shift the magnetization values to

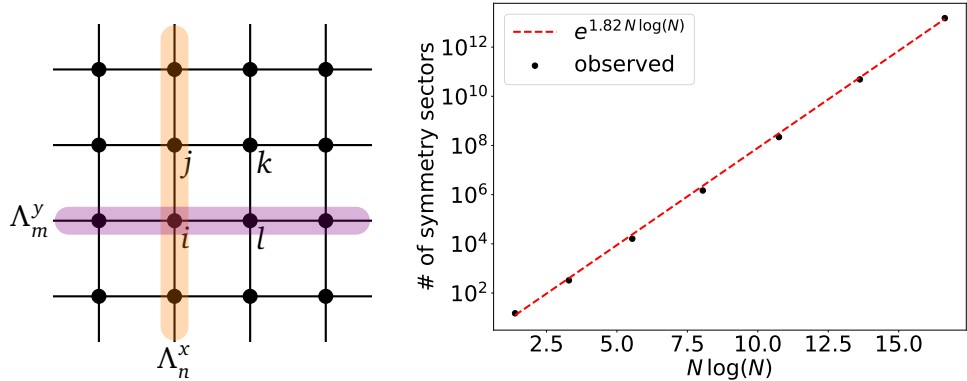

Figure 2: (Left) The ring exchange term acts on elementary plaquettes ($ijkl$). Subsystem symmetry operators $\hat{U}_n^x(\alpha), \hat{U}_m^y(\alpha)$ are defined on columns $\Lambda_n^x$ and rows $\Lambda_m^y$, respectively. (Right) The number of non-empty symmetry sectors for systems of size $N \times N$, for $N = 2, \ldots, 8$.

be integers from 0 to the maximal possible value).

$$
M_n^x = \sum_{i \in \Lambda_n^x} s_i^z + \frac{N_x}{2}, \quad n = 1, \ldots, N_x,
$$

$$
M_m^y = \sum_{i \in \Lambda_m^y} s_i^z + \frac{N_y}{2}, \quad m = 1, \ldots, N_y, \tag{3}
$$

$$
M \equiv \sum_{n=1}^{N_x} M_n^x = \sum_{m=1}^{N_y} M_m^y,
$$

where $s_i^z$ is an eigenvalue of $\hat{S}_i^z$. Let us denote the set of subsystem symmetry quantum numbers as $\mathcal{M} \equiv (M, M^x, M^y) \equiv (M, \{M_n^x\}, \{M_m^y\})$. These quantities remain conserved if we add a term $\hat{H}_z(\{\hat{S}_i^z\})$ arbitrarily dependent on $\hat{S}_i^z$ operators to the Hamiltonian (1).

Additionally, the presence of subsystem symmetries implies the existence of conserved dipole moment in the $x$ and $y$ directions. This follows from the fact that the conserved dipole charge can be expressed in terms of the conserved subsystem charges

$$
D_x = \sum_j j_x s_j^z = \sum_{j_x} j_x M_{j_x}^x,
$$

$$
D_y = \sum_j j_y s_j^z = \sum_{j_y} j_y M_{j_y}^y, \tag{4}
$$

where $j \equiv (j_x, j_y)$ are the coordinates of the lattice sites. The crucial role of dipole conservation and locality in the phenomena of Hilbert space fragmentation has been emphasized in Refs. [23, 24].

The total number of independent subsystem symmetries grows as $N_x + N_y - 1$, linearly with the linear system size, and is sub-extensive, i.e., less than the number of d.o.f., which is $N_x N_y$. Despite an $\mathcal{O}(N_x, N_y)$ number of symmetries, the model is still non-integrable, as we will show in section 5 by analyzing the level spacing statistics. However, the symmetries play a crucial role in the highly constrained dynamics of the model.

We work in the $\hat{S}^z$ basis and denote the local basis states as $|s^z = -1/2\rangle \equiv |0\rangle$, $|s^z = +1/2\rangle \equiv |1\rangle$. Then basis states that span the full Hilbert space can be naturally written as (0,1)-matrices of size $N_x \times N_y$, that is matrices consisting of only 1's and 0's. In this

language, the ring exchange term performs the following local transformation ("flip") of a matrix:

$$\hat{H}_{1\times1}: \begin{pmatrix} 1 & 0 \\ 0 & 1 \end{pmatrix} \longleftrightarrow \begin{pmatrix} 0 & 1 \\ 1 & 0 \end{pmatrix}, \tag{5}$$

while it annihilates all other configurations. The subsystem symmetries imply the conservation of the sum of numbers in each column and row of the matrix. For (0,1)-matrices with prescribed row and column sums there exists the Gale-Ryser algorithm [63–66], which allows to determine whether there exists any matrix satisfying sums $\mathcal{M}$, and in case it does, it allows one to obtain such a matrix. Knowing one such matrix, all other matrices with $\mathcal{M}$ can be obtained by performing non-local flips as in Eq. (5) and combinations thereof [63].

Using the Gale-Ryser algorithm we established the number of non-empty symmetry sectors for $N \times N$ lattices up to $N = 8$ with periodic boundary conditions (p.b.c.) (Fig. 2, right). As one can see, the logarithm of this number is proportional to $N \log(N)$, which coincides with the results from [67], where such proportionality is shown to be typical for continuous subsystem symmetries. This, in particular, shows that subsystem symmetries alone are not responsible for Hilbert space fragmentation that we will see below.

## 3 Fragmentation of the Hilbert space

A combination of the sub-extensive number of conserved charges assigned to 1D subsystems and local plaquette interactions severely fragment the Hilbert space and generically lead to non-ergodic dynamics. In this section we explore kinetic aspects of the Hilbert space structure for the quantum XY-plaquette model.

**A simple illustration:** The phenomenon of Hilbert space fragmentation in the XY-plaquette model can be illustrated as follows. Consider the $4 \times 4$ lattice and let us choose the symmetry sector with quantum numbers $M = 2$, $M^x = (0,1,0,1)$, $M^y = (0,1,0,1)$ (the quantum numbers are counted from left to right and from bottom to top). The Hilbert space of this symmetry sector is spanned by two basis states that can be depicted via the following $4 \times 4$ (0,1)-matrices:

$$A^{(1)} = \begin{pmatrix} 0 & 0 & 0 & 1 \\ 0 & 0 & 0 & 0 \\ 0 & 1 & 0 & 0 \\ 0 & 0 & 0 & 0 \end{pmatrix}, \quad A^{(2)} = \begin{pmatrix} 0 & 1 & 0 & 0 \\ 0 & 0 & 0 & 0 \\ 0 & 0 & 0 & 1 \\ 0 & 0 & 0 & 0 \end{pmatrix}. \tag{6}$$

Although these basis states are in the same symmetry sector, one cannot get from state $A^{(1)}$ to state $A^{(2)}$ by the application of the Hamiltonian in Eq. (1), since both these states have no flippable plaquettes. In other words, $A^{(1)}$ and $A^{(2)}$ are dynamically disconnected from each other. In fact, both $A^{(1)}$ and $A^{(2)}$ are disconnected from any other state and consequently have trivial dynamics under the Hamiltonian (1). Henceforth, we will refer to such completely disconnected states as *inert states*.

**Hilbert space structure:** At the coarsest level, the full Hilbert space $\mathcal{H}$ is decomposed into magnetization sectors $\mathcal{H}_M$. Incorporating subsystem symmetries, each $\mathcal{H}_M$ is decomposed into symmetry sectors $\mathcal{H}_{\mathcal{M}} \equiv \mathcal{H}_{(M,M^x,M^y)}$. Each subsystem symmetry sector can potentially be

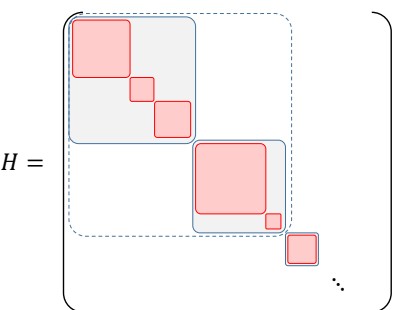

Figure 3: Structure of the block-diagonalized Hamiltonian. Dashed blocks correspond to $M$ values. Solid blue blocks correspond to symmetry sectors characterized by $\mathcal{M} \equiv (M, M^x, M^y)$ quantum numbers. Solid red blocks correspond to fragments. The red block structure is a property of the Hamiltonian alone, while the other blocks are imposed by the symmetries.

decomposed into dynamically disconnected fragments $\mathcal{F}_i^{\mathcal{M}}$.

$$
\begin{aligned}
\mathcal{H} &= \bigoplus_M \mathcal{H}_M = \bigoplus_{\mathcal{M}} \mathcal{H}_{\mathcal{M}}, \\
\mathcal{H}_M &= \bigoplus_{M^x, M^y} \mathcal{H}_{(M, M^x, M^y)}, \\
\mathcal{H}_{\mathcal{M}} &= \bigoplus_i \mathcal{F}_i^{\mathcal{M}}.
\end{aligned}
\tag{7}
$$

The Hamiltonian is then block-diagonalized as shown in the Fig. 3. Note that some fragments or even whole symmetry sectors can be 1-dimensional (inert states). On the other hand, some high-dimensional symmetry sectors might consist only of one fragment. Since the map from subsystem symmetry charges to total dipole moment is many-to-one, block diagonalization based on conserved dipole moment and total charge provides a relatively coarse kinetic description of the Hilbert space.

**Locality and fragmentation:**     In addition to the Hamiltonian $\hat{H}_{1\times1}$, which performs flips on elementary plaquettes in the set $\square_{1\times1}$, we may also consider longer range Hamiltonians that preserve subsystem symmetries (2). Apart from arbitrary $\hat{S}^z$ terms, which are not important for fragmentation, one may consider interactions similar to $\hat{H}_{1\times1}$, but defined on larger rectangles, such as:

$$
\begin{aligned}
\hat{H}_{1\times2}: \quad & \begin{pmatrix} 1 & \bullet & 0 \\ 0 & \bullet & 1 \end{pmatrix} \longleftrightarrow \begin{pmatrix} 0 & \bullet & 1 \\ 1 & \bullet & 0 \end{pmatrix}, \\
\hat{H}_{2\times1}: \quad & \begin{pmatrix} 1 & 0 \\ \bullet & \bullet \\ 0 & 1 \end{pmatrix} \longleftrightarrow \begin{pmatrix} 0 & 1 \\ \bullet & \bullet \\ 1 & 0 \end{pmatrix}, \\
\hat{H}_{2\times2}: \quad & \begin{pmatrix} 1 & \bullet & 0 \\ \bullet & \bullet & \bullet \\ 0 & \bullet & 1 \end{pmatrix} \longleftrightarrow \begin{pmatrix} 0 & \bullet & 1 \\ \bullet & \bullet & \bullet \\ 1 & \bullet & 0 \end{pmatrix}, \quad \text{etc.},
\end{aligned}
\tag{8}
$$

where $\bullet$ denotes an arbitrary spin not taking part in the interaction. For later convenience we introduce three Hamiltonians: $\hat{H} \equiv \hat{H}_{1\times1}$, $\hat{H}' \equiv \hat{H}_{1\times1} + \hat{H}_{1\times2} + \hat{H}_{2\times1}$ and $\hat{H}'' \equiv \hat{H}_{1\times1} + \hat{H}_{1\times2} + \hat{H}_{2\times1} + \hat{H}_{2\times2}$.

In addition to the terms that flip spins at the corners of a rectangle, an interaction that preserves subsystem symmetries (2) generically would consist of spin-raising and spin-lowering

operators arranged in an alternating way at the corners of a figure of an arbitrary shape composed of alternating vertical and horizontal edges. More elaborate examples of such figures are:

$$
\hat{H}_{\square}: \quad
\begin{pmatrix} 1 & \bullet & 0 \\ 0 & 1 & \bullet \\ \bullet & 0 & 1 \end{pmatrix}
\longleftrightarrow
\begin{pmatrix} 0 & \bullet & 1 \\ 1 & 0 & \bullet \\ \bullet & 1 & 0 \end{pmatrix},
$$

$$
\hat{H}_{\square}: \quad
\begin{pmatrix} \bullet & 1 & 0 & \bullet \\ \bullet & \bullet & 1 & 0 \\ 1 & 0 & \bullet & \bullet \\ 0 & \bullet & \bullet & 1 \end{pmatrix}
\longleftrightarrow
\begin{pmatrix} \bullet & 0 & 1 & \bullet \\ \bullet & \bullet & 0 & 1 \\ 0 & 1 & \bullet & \bullet \\ 1 & \bullet & \bullet & 0 \end{pmatrix},
\tag{9}
$$

$$
\hat{H}_{\square}: \quad
\begin{pmatrix} \bullet & 1 & \bullet & 0 \\ \bullet & \bullet & \bullet & \bullet \\ 0 & \bullet & \bullet & 1 \\ 1 & 0 & \bullet & \bullet \end{pmatrix}
\longleftrightarrow
\begin{pmatrix} \bullet & 0 & \bullet & 1 \\ \bullet & \bullet & \bullet & \bullet \\ 1 & \bullet & \bullet & 0 \\ 0 & 1 & \bullet & \bullet \end{pmatrix}.
$$

It is easy to see that the magnetization in each row and column is conserved, since each edge of the figure is either vertical or horizontal and contains exactly one 0 and one 1, which exchange positions under the Hamiltonian action. This process does not change row and column sums. Note that the boundaries of the figures can also be self-intersecting, as shown in the third example in Eq. (9).

In Appendix G, we discuss the role of the ring-exchange interaction "shape" for the Hilbert space fragmentation in systems with subsystem symmetries, and its interplay with the locality of the interaction.

**Counting inert states:** The number of inert states can be calculated using the transfer matrix approach, analogous to the construction of inert states in a 1D dipole-conserving spin-1 chain in Ref. [23, 24]. The process of construction resembles the notion of mathematical induction. First, we consider a lattice of size $N_x \times N_y$ with p.b.c. in the $x$-direction, and assume that the system is in an inert state. It is thus in one of the product states in $\hat{S}^z$-basis and can be represented by a (0,1)-matrix. Now, we would like to stack an additional $(N_y + 1)$-th row on top, such that the system remains in an inert state. Allowed combinations of 0's and 1's in the $(N_y + 1)$-th row depend on the configuration in the $N_y$-th row. We can construct a matrix $T$ depicting which configurations in rows $N_y$ and $N_y + 1$ are compatible with each other. Let us give an example of such a matrix for the case of $N_x = 3$:

$$
T = \begin{pmatrix}
 & {\scriptstyle(000)} & {\scriptstyle(001)} & {\scriptstyle(010)} & {\scriptstyle(011)} & {\scriptstyle(100)} & {\scriptstyle(101)} & {\scriptstyle(110)} & {\scriptstyle(111)} & \\
 & 1 & 1 & 1 & 1 & 1 & 1 & 1 & 1 & {\scriptstyle(000)} \\
 & 1 & 1 & 0 & 1 & 0 & 1 & 0 & 1 & {\scriptstyle(001)} \\
 & 1 & 0 & 1 & 1 & 0 & 0 & 1 & 1 & {\scriptstyle(010)} \\
 & 1 & 1 & 1 & 1 & 0 & 0 & 0 & 1 & {\scriptstyle(011)} \\
 & 1 & 0 & 0 & 0 & 1 & 1 & 1 & 1 & {\scriptstyle(100)} \\
 & 1 & 1 & 0 & 0 & 1 & 1 & 0 & 1 & {\scriptstyle(101)} \\
 & 1 & 0 & 1 & 0 & 1 & 0 & 1 & 1 & {\scriptstyle(110)} \\
 & 1 & 1 & 1 & 1 & 1 & 1 & 1 & 1 & {\scriptstyle(111)}
\end{pmatrix},
\tag{10}
$$

where rows and columns correspond to the 8 possible configurations of the $N_y$'th and $(N_y+1)$'th rows of the lattice, respectively. We can then write

$$
\begin{pmatrix} N_{(000)} \\ \vdots \\ N_{(111)} \end{pmatrix}_{(N_y+1)} = T \begin{pmatrix} N_{(000)} \\ \vdots \\ N_{(111)} \end{pmatrix}_{(N_y)},
\tag{11}
$$

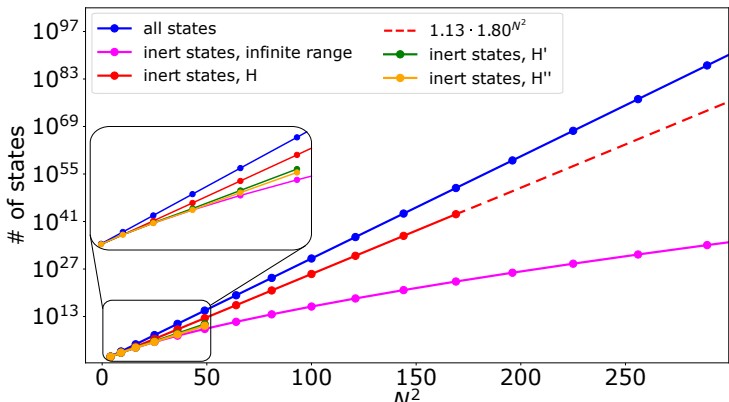

Figure 4: The number of inert states on $N \times N$ lattices for different ranges of rectangular interactions: (red) $\hat{H}$, (green) $\hat{H}'$, (yellow) $\hat{H}''$, (pink) infinite range (rectangles of any size), and, for comparison, (blue) the full dimensionality of the Hilbert space: $2^{N^2}$. Red curve scales as $1.13 \cdot 1.80^{N^2}$ (a straight line on the log plot). Green and yellow curves have slight deviations from straight lines. However, due to the limited number of data points, it is hard to tell if they grow exponentially or sub-exponentially. For the system with no Hilbert space fragmentation (i.e., infinite range rectangular interactions) the number of inert states grows sub-exponentially (pink). An analytic expression for the pink curve was obtained using results from Ref. [68].

where $\left(N_{(000)}, \dots, N_{(111)}\right)_{(N_y)}$ is a vector depicting the number of inert configurations on a lattice of length $N_y$ in the $y$-direction that end with a row $(000), \dots, (111)$, correspondingly. Using this recursive relation, we can calculate the number of inert states on the $N_x \times N_y$ lattice as $\mathbf{v}^T T^{N_y} \mathbf{v}$ with $\mathbf{v} = (1, 1, \dots, 1)^T$ if we assume open boundary conditions (o.b.c.) in the $y$-direction, and as $\text{Tr}\left(T^{N_y}\right)$ if we assume p.b.c. in the $y$-direction.

One can as well calculate the exact number of inert states for longer-range interactions using a similar approach, where instead of matrix $T$ one has to construct a higher-dimensional tensor of size $2^{N_x} \times \cdots \times 2^{N_x}$ and then take an appropriate contraction of $N_y$ such tensors.

In Fig. 4 we compare how the number of inert states grows with the number of d.o.f. for lattices $N \times N$ for different interaction ranges. As expected, increasing interaction range leads to the decrease in the number of inert states. The minimal possible number of inert states is

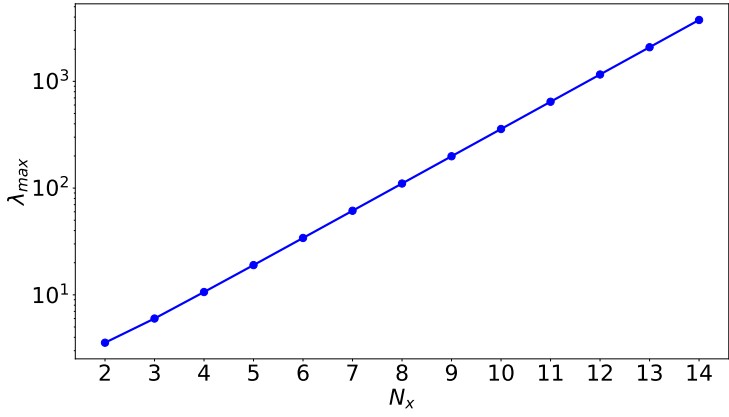

Figure 5: Scaling of the largest eigenvalue $\lambda_{\max}$ of the transfer matrix $T$, depending on the lattice size in $x$-direction $N_x$. The fitted line is $\lambda_{\max} \approx 1.80^{N_x}$.

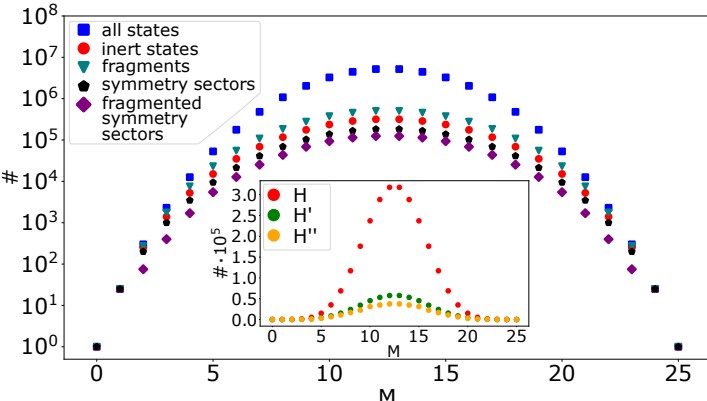

Figure 6: $M$-dependence of (blue) the total number of states, (red) the number of inert states, (turquoise) the number of fragments, (black) the number of symmetry sectors, (purple) the number of fragmented (i.e., containing more than one fragment) symmetry sectors, for the $5 \times 5$ lattice with p.b.c. and the Hamiltonian $\hat{H}$. Inset: $M$-dependence of the number of inert states for different (rectangular) interaction ranges: (red) $\hat{H}$, (green) $\hat{H}'$, (yellow) $\hat{H}''$.

dictated by the subsystem symmetries, and is achieved when the interaction range is increased to infinity (pink curve). In this case every inert state constitutes a 1-dimensional symmetry sector. The red straight line in Fig. 4 indicates that local $1 \times 1$ interactions lead to exponential (in the number of d.o.f.) growth of the inert subspace. More precisely, fitting the results for $N = 2, \ldots, 13$, we get that the number of inert states scales as $\propto 1.80^{N^2}$. In addition, we can estimate this scaling behavior in another way. Since $\mathrm{Tr}(T^N) = \lambda_1^N + \ldots + \lambda_{2^N}^N$, where $\lambda_i$ are the eigenvalues of $T$, in the thermodynamic limit this sum is dominated by the largest eigenvalue $\lambda_{\max}$ and therefore the number of inert states in a system $N_x \times N_y$, $N_y \to \infty$, scales as $(\lambda_{\max}(N_x))^{N_y}$. In Fig. 5 we plot the largest eigenvalues for $N_x = 1, \ldots, 13$ and determine that it scales as $\propto 1.80^{N_x}$, leading to $\propto 1.80^{N_x N_y}$ scaling of the total number of inert states when both $N_x, N_y \to \infty$, which coincides with our first estimation obtained from fitting the exact results in Fig. 4.

The extensive number of inert states demonstrates "shattering" of the Hilbert space. It should be noted that shattering is not simply a consequence of the sub-extensive number of conserved quantities due to subsystem symmetries, since upon increasing the interaction range to infinity the dimension of the inert subspace no longer grows exponentially, as shown in pink. The analytic expression for the pink curve was obtained in Ref. [68]:

$$\sum_{j=1}^{N} (-1)^{N+j} j! \begin{Bmatrix} N \\ j \end{Bmatrix} (j+1)^N < \frac{1}{2} N^{2N} e^N \tag{12}$$

at large $N$, where $\begin{Bmatrix} N \\ j \end{Bmatrix}$ is the Stirling number of the second kind (the bound is calculated in Appendix F). The quantity scales slower than $e^{N^2}$ and therefore we conclude that in case of the "rectangular" interactions the locality of interaction plays a crucial role for the extensive fragmentation of the Hilbert space.

**Distribution of fragment sizes:** Here we present numerical results revealing the detailed structure of the Hilbert space for $N \times N$ lattices with p.b.c., for $N$ up to 5 for the Hamiltonians $\hat{H}, \hat{H}'$, and $\hat{H}''$. We note that, since the largest lattice size we consider is $5 \times 5$, the rectangular interactions of length up to $2 \times 2$ completely remove the Hilbert space fragmentation, since they act on roughly the half of the lattice size. Our numerical results confirm this intuition.

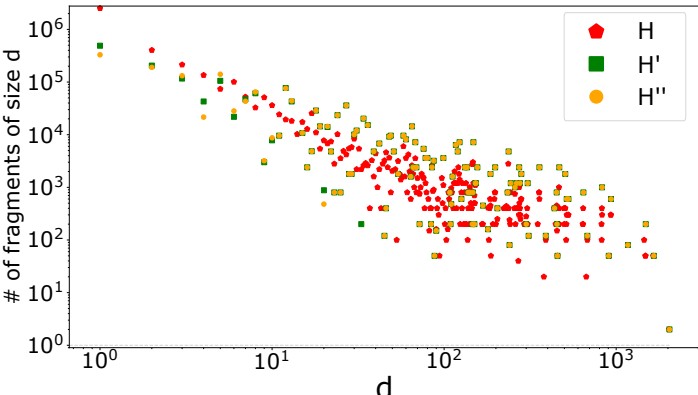

Figure 7: Fragment size distribution for the $5 \times 5$ lattice, for different interaction ranges.

In order to reveal how the symmetry sectors split into disconnected fragments, we recursively iterate over basis states and apply the corresponding Hamiltonian.

In Fig. 6 we plot the dependence of the numbers of inert states, fragments, symmetry sectors and fragmented symmetry sectors on the total magnetization $M$. We see that the "typical" states lie in sectors with magnetization close to 0 (the middle of the range in $M$) and that all magnetization sectors (except for the extremal ones) exhibit extensive fragmentation with exponentially many inert states and fragmented symmetry sectors. The reduction of the number of inert states upon the increase of the interaction range is seen in the inset of Fig. 6.

It is also instructive to look at the size distribution of fragments. In Fig. 7 this data is plotted for the $5 \times 5$ lattice. We observe a polynomial decrease (a straight line on the log-log plot) in the amount of fragments as their dimensionality increases. While as the range of interaction increases, the number of small-dimensional fragments decreases, and the number of large-dimensional fragments increases. It is also notable that even for $\hat{H}_{1\times1}$ interactions there exist large fragments (for the $5 \times 5$ lattice: 2 fragments of size 2030). We will analyze the thermal properties of these large fragments in some detail in the next section.

We further study the connection between magnetization sector and largest fragment sizes. Figure 8 shows the size of the largest symmetry sector (red) and the largest fragment (blue) as a function of $M$ for lattices $4 \times 4$ and $5 \times 5$, respectively. These plots immediately reveal the qualitative difference between even×even and odd×odd lattices: when $N = 2n$, the largest symmetry sector and the largest fragment lie in the mid-magnetization sector, or more precisely in the symmetry sector $M^x = M^y = (n, n, \ldots, n)$, i.e., the magnetization is $n$ in each column and row; when $N = 2n+1$, the largest symmetry sector and the largest fragment lie in symmetry sectors $M^x = M^y = (n+1, n+1, \ldots, n+1)$ and $M^x = M^y = (n, n, \ldots, n)$. Another interesting feature one can notice is that while the largest fragment occupies almost the whole symmetry sector in most of the magnetization sectors, there are magnetization sectors around $M = N, N^2 - N$, with a marked deviation from such behavior. This indicates that symmetry sectors in these magnetization sectors are fragmented into several fragments of comparable sizes, rather than into one large and several small fragments. This statement is further supported by Fig. 9, where the largest number of fragments in a single symmetry sector for different magnetization sectors have been plotted. The most fragmented symmetry sectors turn out to be $M^x = M^y = (1, 1, \ldots, 1)$ and $M^x = M^y = (N-1, N-1, \ldots, N-1)$. As can be seen, there is no fragmentation for the Hamiltonian $\hat{H}''$ which includes terms up to $2 \times 2$ interaction range, while there are still some remnants of fragmentation for the Hamiltonian $\hat{H}'$ with $1 \times 2$ and $2 \times 1$ interaction range.

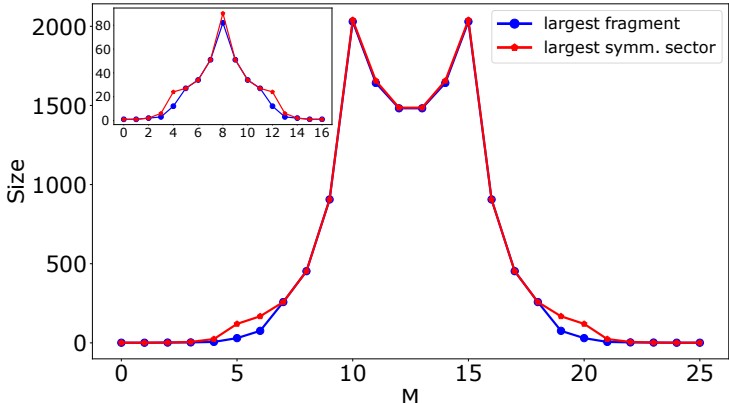

Figure 8: (Blue) the size of the largest fragment in each $M$-sector; (Red) the size of the largest symmetry sector in each $M$-sector, for the Hamiltonian $\hat{H}$ ($1 \times 1$ interaction range) on the $5 \times 5$ lattice with p.b.c. Inset: same for the $4 \times 4$ lattice with p.b.c.

The size of a given fragment can be deduced from the time evolution of the wave function initialized in a pure state $|\psi_0\rangle$ belonging to this fragment. In particular, one can look at the Fourier transform (FT) of the Loschmidt echo, $|\langle\psi(t)|\psi_0\rangle|^2$, and read off the size of the fragment from the number of peaks in the FT spectrum (see Appendix B for details). Since the positions of peaks depend on the energy differences in the energy spectrum of the fragment, one might apply random magnetic fields in the $z$-direction in order to avoid the overlap of the peaks.

## 4 Shielding structures

It can happen that while a state is not inert, some specific spins might still remain inert throughout the time evolution of the system. For instance, if a particular spin $\hat{S}_i$ has the same $s_i^z$ eigenvalue in all the basis states of a certain fragment of the Hilbert space, this spin is ensured to have a fixed (non-evolving) $s_i^z$ eigenvalue as long as the system is in a pure or mixed state belonging to this fragment. More generally, a spin $\hat{S}_i$ will remain inert under the time evolution of a system initialized in a density matrix with support restricted to fragments where all basis states have the same value of $s_i^z$. We will show that it is possible to construct "shielding structures": sets of inert spins separating regions of non-inert spins ("active regions"). This essentially leads to breaking up the system into spatial regions that are completely disentangled from each other and can therefore be considered as disconnected systems with respect to time evolution, and thermal or quantum fluctuations.

Now, let us deduce the conditions for a spin to be inert. For simplicity, here we will consider only the interaction encoded in $\hat{H}$ (we discuss longer range interactions in the Appendix A). We assume the system is in one of the $\hat{S}^z$-basis states which we write in (0,1)-matrix notation (generalization to eigenstates or mixed states is straightforward). Whether the chosen spin belongs to a flippable plaquette or not (i.e., whether it is "immediately flippable" by the action of the Hamiltonian), depends on its immediate neighborhood that contains eight surrounding spins. In Fig. 10 we list all sufficient local configurations of spins (in black) that make it impossible to immediately flip the central spin (denoted in red).

Working in the $\hat{S}_z$ basis, we will be coloring all immediately non-flippable spins in red. By the above discussion, the neighborhood of each of these red spins contains at least one of the 22 configurations in Fig. 10 that protect it from immediate flipping. If we can find a subset

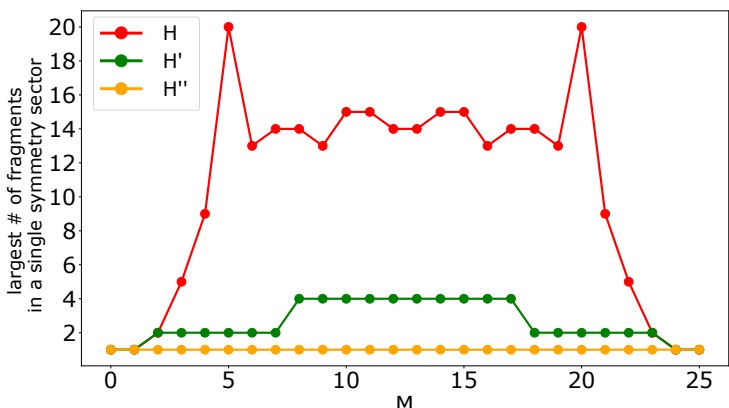

Figure 9: Lattice $5 \times 5$ with p.b.c.; The largest number of fragments in a single symmetry sector for each $M$-sector.

of red spins $A_s$ such that the neighborhood of each of the spins in $A_s$ contains at least one of above structures that is itself in $A_s$, then $A_s$ is a shielding structure that is inert at all times. It is now straightforward to build a shielding structure starting from one spin and following the iterative procedure, as illustrated in Fig. 11. We start with a single spin and ensure it is immediately non-flippable (i.e., red) by surrounding it with one of the local configurations in Fig. 10. Now we do the same for newly added spins. We then iteratively repeat this procedure, gradually building up the shielding structure (as exemplified in Fig. 11).

Three examples of such inert shielding structures are presented in Fig. 12. The elementary building blocks of the shields, as one can see from the Fig. 10, are: the straight horizontal and vertical lines, "4-junctions" (intersections of four diagonal lines) and two types of "3-junctions". These building blocks define the "branching rules" of shielding structures. One verifies from the shape of these elementary building blocks that in order for a shield to close onto itself, it has to go through the p.b.c. (or terminate at the o.b.c.), meaning that shields are

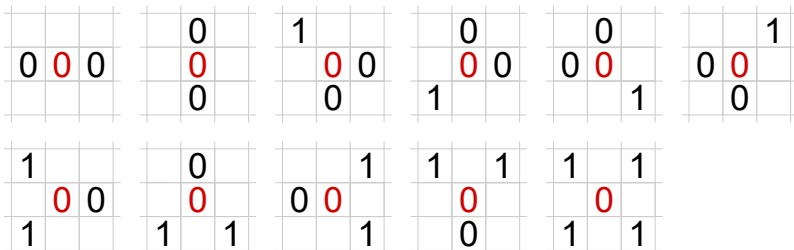

Figure 10: If a spin in red is surrounded by one of the presented local configurations of spins in black, it cannot be immediately flipped. Only half of the configurations is shown. The second half is obtained by a global spin flip, $0 \leftrightarrow 1$ (22 configurations in total).

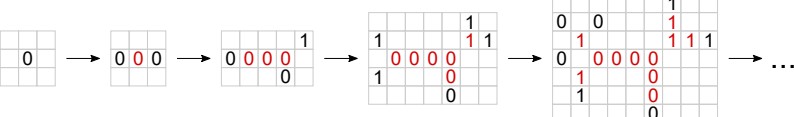

Figure 11: An example of a shielding structure construction. We iteratively ensure inertness of each spin by adding new appropriate spins in its surrounding. This is repeated until the structure "closes onto itself".

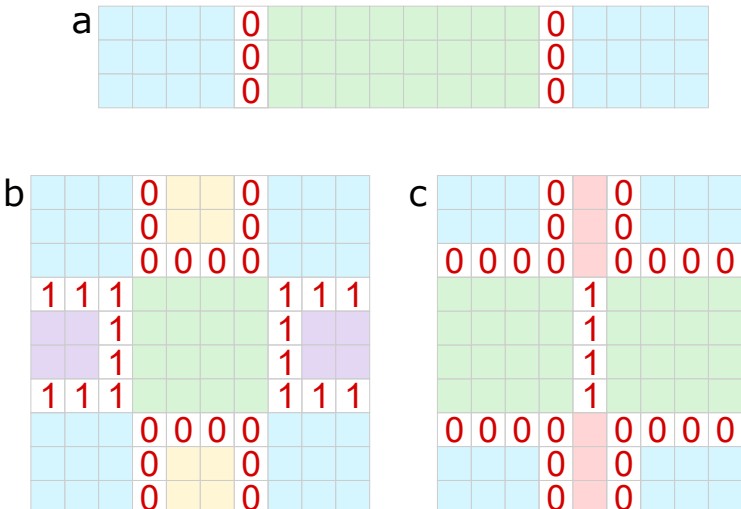

Figure 12: Examples of shielding structures separating the system into spatially disconnected disentangled active regions (painted in different colors); p.b.c are assumed. (a) Two simple straight shields can separate the system into two active regions. (b, c) More complicated shielding structures, including "3-junctions". In (c), the spins in the region painted in solid red color can be chosen arbitrarily, but nevertheless will necessarily remain inert.

necessarily *non-local* objects. Non-locality of the shields implies that in the thermodynamic limit the probability of having a shield in the system tends to zero. The isolated active regions, however, can be localized in space, as presented in Fig. 12.

In Ref. [23], it was pointed out that for a generic dipole conserving system, shielding structures isolating an active region should be at least of the size of the active region. In the particular case of the XY-plaquette model, however, the thickness of the shields depends only on the interaction range, while the separated active regions can be arbitrarily large. For example, for $1 \times 1$ interaction it suffices for the shields to be only 1 site thick. For $1 \times 2, 2 \times 1$ and $2 \times 2$ interactions, the 2-site thickness of the shields will be enough, and so on: for $n \times m$ interactions, shields of thickness $\max(n, m)$ will be able to prevent active regions from thermalizing with each other.

Existence of such shielding structures significantly affects the correlation function properties, such as, for instance, the equal-time spin-spin correlation function $C_{0i}^z(t) \equiv \langle \hat{S}_0^z(t) \hat{S}_i^z(t) \rangle - \langle \hat{S}_0^z(t) \rangle \langle \hat{S}_i^z(t) \rangle$. Generically, shields do not possess translational or rotational symmetry, and since they split the system into disentangled spatial regions, $C_{0i}^z(t)$ also loses translational and rotational invariance and becomes crucially dependent on the initial state and the choice of the coordinate origin $(j_x, j_y) = (0, 0)$. $C_{0i}^z(t)$ can be non-zero only if the corresponding spins at sites 0 and $i$ belong to the same active region. Otherwise, if $i$ lies in a different active region, separated from the origin by shields, $C_{0i}^z(t) = 0$ at all times $t$. We confirm this with numerical calculations. We initialize the system initialized in the state

$$\begin{pmatrix} 0 & 1 & 0 & 1 & 0 & 1 & 0 & 1 \\ 0 & 0 & 1 & 0 & 0 & 0 & 1 & 0 \\ 0 & 1 & 0 & 1 & 0 & 1 & 0 & 1 \\ 0 & 0 & 0 & 0 & 0 & 0 & 0 & 0 \end{pmatrix}, \tag{13}$$

which has two active regions (in black) separated by shields (in red). In Fig. 13 we compare the time evolution of $C_{0i}^z(t)$ for two different choices of the origin (denoted with a black dot).

One can see that the correlation function has non-trivial dynamics only when both spins lie within the same active region, and is equal to zero when spins lie in different active regions or one of the spins belongs to the shielding structure (and thus fully inert). The unusual behavior of the correlation function in the models with rectangular ring-exchange interactions was investigated in the quantum circuit setting in Ref. [69].

## 5 Entanglement entropy

**Entanglement entropy of eigenstates:**    As we have shown in the previous sections, subsystem symmetries lead to an extensive number of disconnected components of the Hilbert space, while local ring-exchange interactions fragment these components even further, leading, in particular, to an exponential number of inert states. One way to explicitly show that such fragmentation of the Hilbert space leads to the violation of ETH is to probe the entanglement entropy.

We divide the system into two parts, $A$ and $B$ with a straight cut through the middle (or, for $(2n+1) \times (2n+1)$ lattices, approximately the middle) of the lattice. The Von Neumann bipartite entanglement entropy is defined as

$$S_{AB}(\rho_A) = -\text{Tr}(\rho_A \ln \rho_A), \tag{14}$$

where $\rho_A = \text{Tr}_B \rho$ is the reduced density matrix of the full density matrix $\rho$ and the trace $\text{Tr}_B$ is taken over the $B$ subregion.

In Fig. 14 we plot the entanglement entropy of eigenstates as a function of their eigenenergy for the Hamiltonian $\hat{H}$. In contrast to systems obeying ETH, where states in the middle of the energy spectrum are highly entangled, here we observe the coexistence of states ranging from low-entangled to high-entangled ones for practically any narrow energy window. In this plot, all inert states fall into one point $(E = 0, S_{AB} = 0)$, while the visible line $(E, S_{AB} = 0)$ at the bottom of the plot corresponds to states in fragments where the bipartition cut separates disconnected active regions (i.e., there are shielding structures separating A and B regions), and therefore the entanglement between A and B is zero.

**Zero energy eigenstates:**    Another prominent feature is the vertical line of states at $(E = 0, S_{AB})$. Let us explain the abundance of the zero-energy eigenstates. In our choice of basis, each block of the full Hamiltonian $\hat{H}$ (corresponding to a fragment $\mathcal{F}_i$), $\hat{H}_{\mathcal{F}_i}$, is represented by a symmetric (0,1)-matrix with zeros on the main diagonal. 1's in this Hamiltonian essentially show which basis states can be connected via a single plaquette flip. We can illustrate this with a connectivity graph in Hilbert space; examples are presented in Fig. 15, where (a), (b) and (c) graphs correspond to the Hamiltonians

$$\hat{H}_{\mathcal{F}_i}^{(a)} = \begin{pmatrix} 0 & 1 & 1 \\ 1 & 0 & 0 \\ 1 & 0 & 0 \end{pmatrix}, \ \hat{H}_{\mathcal{F}_i}^{(b)} = \begin{pmatrix} 0 & 1 & 1 & 1 \\ 1 & 0 & 0 & 0 \\ 1 & 0 & 0 & 0 \\ 1 & 0 & 0 & 0 \end{pmatrix},$$

$$\hat{H}_{\mathcal{F}_i}^{(c)} = \begin{pmatrix} 0 & 1 & 0 & 0 \\ 1 & 0 & 1 & 0 \\ 0 & 1 & 0 & 1 \\ 0 & 0 & 1 & 0 \end{pmatrix}, \tag{15}$$

respectively. Recall that if a matrix $n \times n$ has rank $m \leq n$, it has exactly $n - m$ zero eigenvalues. The rank $m$ is strictly less than $n$ when the columns (rows) are linearly dependent, which for a (0,1)-matrix means that some of the columns (rows) are identical, or that there is a

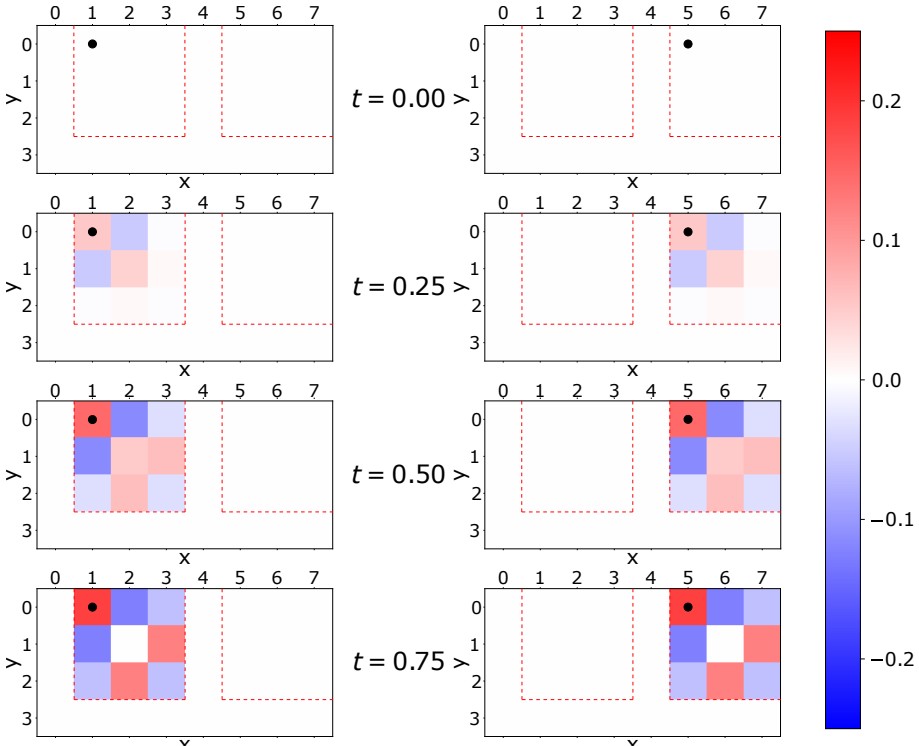

Figure 13: Time evolution of the equal-time spin-spin correlation function $C_{0i}^{z}(t)$, with $i \equiv (j_x, j_y)$ and the origin chosen at the black dot, from the initial state (13). The initial state contains shielding structures that split the system into two active regions (denoted by red dashed lines). Two choices of the origin are compared (left and right columns). The correlation function is non-zero only in the active region containing the origin.

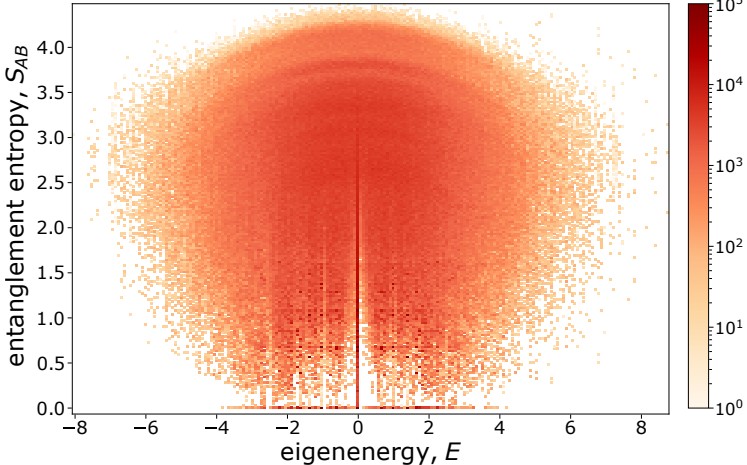

Figure 14: Bipartite entanglement entropy of eigenstates as a function of their eigenenergy, for the entire spectrum of the model $\hat{H}$ on the $5 \times 5$ lattice. The color denotes the number of states with a particular $S_{AB}$ and $E$.

zero column (row). In terms of the connectivity graphs, this corresponds to a permutation symmetry of the graph, i.e., there are states upon permutation of which the graph remains the same [1]. For instance, in Fig. 15(a), the permutation of states 2 and 3 does not change the graph. The same holds true for Fig. 15(b) with regards to the permutation of states 2,3,4. In the corresponding Hamiltonians this can be seen as identical columns (rows), and thus the fragment in (a) has a zero energy state and the fragment in (b) has a doubly degenerate zero energy state. The connectivity graph in Fig. 15(c) does not possess any permutation symmetries and therefore the corresponding fragment has no zero energy level. Owing to this permutation symmetry of the basis states, one obtains several zero energy eigenstates as can be in Fig. 14.

**Energy and entanglement properties of large fragments:**     In this subsection we numerically investigate integrability properties of the model. For this, we resort to large fragments and study their energy level spacing statistics and the spectrum of the entanglement entropy of eigenstates. As an example, we choose a 9856-dimensional fragment lying in a symmetry sector with quantum numbers $M = 11, M^x = (1, 3, 2, 1, 2, 2), M^y = (2, 2, 2, 2, 2, 1)$ of the 6×6 lattice with p.b.c. We consider a one-parameter family of Hamiltonians with random magnetic field,

$$\hat{H}_{\text{rand}}(x) = (1 - x)\hat{H}_{1\times 1} + x \sum_i h_i \hat{S}_i^z, \tag{16}$$

where $x \in [0, 1]$ and $h_i$ is a random number uniformly distributed between -1 and 1. Note that the second term preserves the subsystem symmetries.

We denote the difference between consecutive energy levels as $s_n = E_{n+1} - E_n$ and introduce the ratio of consecutive level spacings, $r = s_n/s_{n-1}$, as well as the value $\tilde{r}_n = \min(s_n, s_{n-1})/\max(s_n, s_{n-1})$. The distribution of ratios $r$ in Fig. 16 shows that in the absence of disorder ($x = 0$), the spacing statistics is Poisson and the model appears integrable (at least inside of the considered fragment). However, upon the addition of small disorder, $x = 0.01$, the statistics changes to Wigner-Dyson, with level repulsion that can be approximated by the corresponding function for the Gaussian orthonormal ensemble (GOE) [70] – the random matrix theory result that indicates non-integrability. The average value of $\tilde{r}$ changes from $\langle \tilde{r} \rangle \approx 0.3890$, which is close to the $\tilde{r}$−value for the Poisson statistics, $\langle \tilde{r} \rangle_{\text{Poisson}} \approx 0.3863$, to $\langle \tilde{r} \rangle \approx 0.5165$ that is closer to $\langle \tilde{r} \rangle_{\text{GOE}} \approx 0.5359$.

Note that the random magnetic field along $z$-direction does not violate subsystem symmetries, nor does it alter any fragmentation properties. Therefore, for any value of $x$, the model is non-ergodic, since it exhibits an abundance of low-dimensional fragments, including an exponential number of inert states. Thus, at low values of $x$ we have an example of a non-integrable model that violates ETH.

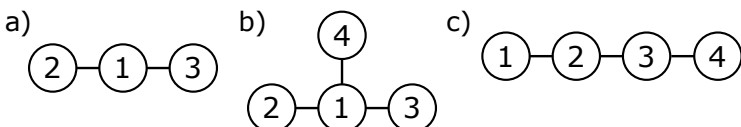

Figure 15: Examples of connectivity graphs in $\hat{S}^z$-basis for different fragments, corresponding to the Hamiltonians (15). Graphs (a) and (b) possess permutation symmetry: graph (a) does not change upon the exchange of states 2 and 3, (b) does not change upon any permutation of states 2,3,4. As a result, the corresponding Hamiltonians contain (a) two, (b) three equal columns (rows), which results in them having (a) one-, (b) 2-dimensional eigenspace with energy zero.

---

[1]Unless the symmetric states are connected to each other, as in the case with a fully-connected graph of 3 states.

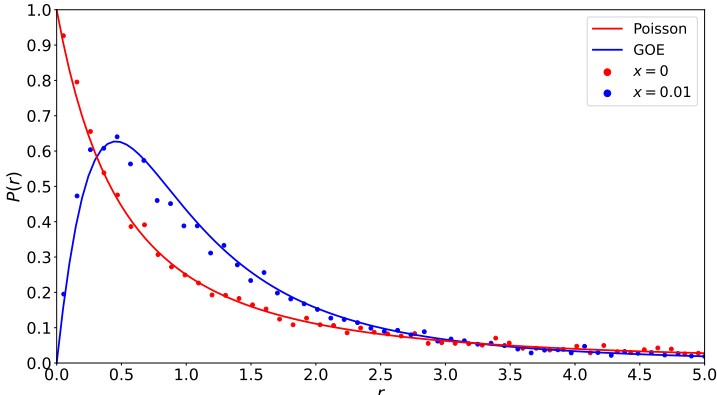

Figure 16: Probability distribution of the ratio $r$ of consecutive energy level spacings, $r_n = (E_{n+1} - E_n)/(E_n - E_{n-1})$, for a large (9856-dimensional) Hilbert space fragment of the model (16) on a $6 \times 6$ lattice with p.b.c, for two values of parameter $x$: (red) no disorder, $x = 0$, (blue) small disorder, $x = 0.01$. Solid lines correspond to distributions of $r$ for (red) Poisson ensemble, $P(r) = (1+r)^{-2}$, (blue) GOE, $P(r) = \frac{27}{8} \frac{r+r^2}{(1+r+r^2)^{5/2}}$ (surmise from Ref. [70]).

At $x = 0$, while being integrable, the considered fragment nevertheless has only high-entangled eigenstates in the middle of the spectrum (red arc in Fig. 17), indicating that the system thermalizes well within the fragment. However, at $x = 0.01$ scar states with low entanglement appear (blue dots in Fig. 17). Upon further increase of $x$, the entanglement entropy of these scar states increases until they eventually merge with the arc. We also note that at high values of $x \sim 0.58$, the GOE spacing distribution gradually changes back to Poisson, while all eigenstates become low-entangled. Whether this transition happens at $x < 1$ in the thermodynamic limit, is an open question. We defer studies of this transition and more precise characterization of the transition in the vicinity of $x = 0$ to later work.

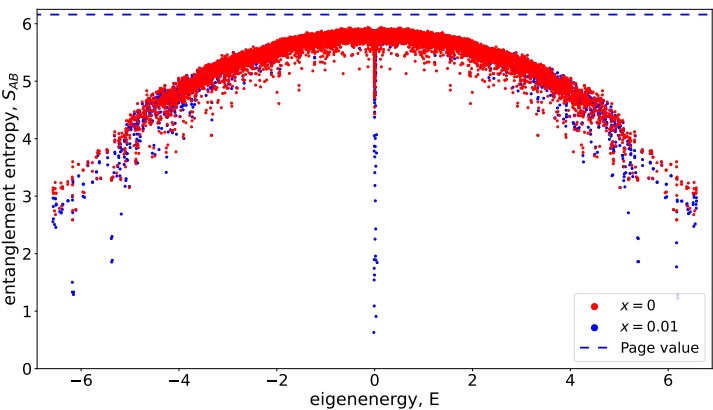

Figure 17: Bipartite entanglement entropy of eigenstates as a function of their eigenenergy, for a large Hilbert space fragment of the model (16) on the $6 \times 6$ lattice with p.b.c, for (red) $x = 0$, (blue) $x = 0.01$. Dashed line: Page value for this fragment, $S_{AB,\text{Page}} \approx 6.16$. With no disorder, the fragment appears thermalizing, since the entanglement entropy of eigenstates form an arc with high-entangled states in the middle of the spectrum (with some outlying states, which are nevertheless have high entanglement). However, upon adding small disorder, low-entangled scar states appear.

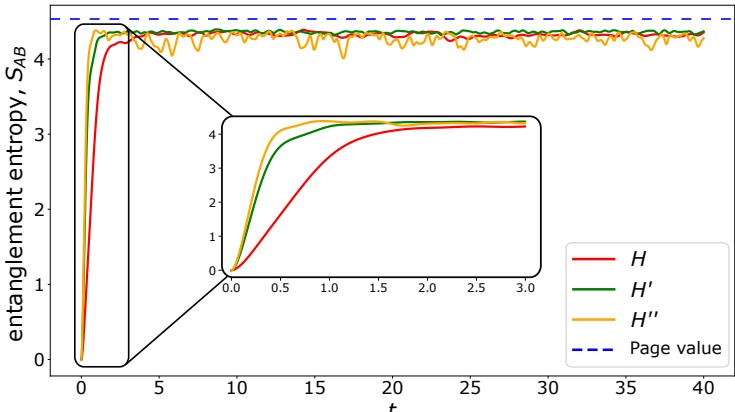

Figure 18: Growth of entanglement entropy $S_{AB}$ for an initial random state in $S^z$-basis in the largest symmetry sector of the $5 \times 5$ lattice with p.b.c., for different interaction ranges. $S_{AB}$ saturates to the value $S_{AB} \approx 4.36$ (independent of the interaction range) below the Page value for this symmetry sector, $S_{AB,\text{Page}} \approx 4.53$, at a rate that increases upon the increase of the interaction range.

**Time evolution of entanglement entropy:** Another way to probe thermalization is to look at how entanglement between subregions A and B grows in time. We calculate the entanglement entropy $S_{AB}(t)$ as a function of time, initializing the system in different fragments. For sufficiently low-dimensional fragments, the entanglement entropy evolves periodically in time, with perfect recurrences occurring simultaneously with the recurrences in the Loschmidt echo. In contrast, for larger fragments, $S_{AB}$ is expected to saturate at some value. We first examine the largest symmetry sector of the $5 \times 5$ lattice. This symmetry sector consists of a large fragment of dimension 2030 and 10 inert states when considering $\hat{H}$. Increasing the interaction range has two effects: (i) the inert states merge into the largest fragment, and (ii) the connectivity of the basis states increases (in the sense that more links appear on the connectivity graph introduced above), which leads to faster thermalization. This can be seen in Fig. 18, where we plot $S_{AB}$ depending on time $t$ after initializing the system in one of the $\hat{S}^z$ basis states. The entanglement entropy saturates to a value $S_{AB} \approx 4.36$ that is slightly lower than the maximal possible (Page) value for the corresponding symmetry sector, $S_{AB,\text{Page}} \approx 4.53$ [2]. The Page value has been evaluated through the exact formula $S_{AB,\text{Page}} = \sum_{k=n+1}^{mn} \frac{1}{k} - \frac{m-1}{2n}$, where $m = \min(d_A, d_B)$, $n = \max(d_A, d_B)$ [71], and $d_{A(B)}$ is the Hilbert space dimension of the subsystem A(B) in the considered symmetry sector.

From Fig. 18, it is clear that upon the increase of the interaction range the saturation rate of the entanglement entropy increases, i.e., entanglement between A and B grows faster. However, for this particular case the saturation value of $S_{AB}$ is independent of the interaction range.

We can also find a symmetry sector where the saturation value crucially depends on the interaction range. From Figs. 8 and 9, we know that symmetry sectors with $M = N$, $M^x = M^y = (1, 1, \ldots, 1)$ and $M = N^2 - N$, $M^x = M^y = (N-1, N-1, \ldots, N-1)$ consist of several fragments comparable in size. Therefore, we expect that upon the increase of the interaction range, the "available" Hilbert space will substantially increase. In Fig. 19, we plot $S_{AB}(t)$ for such a symmetry sector on a $6 \times 6$ lattice with o.b.c. (on the $5 \times 5$ lattice the dimensions of separate fragments in such a sector are too small to exhibit saturation). This symmetry sector has dimension 720 and under $\hat{H}$ is fragmented into 258 fragments with sizes ranging

---

[2]The fact that entanglement entropy does not reach the Page value is consistent with the fact that ETH is expected to hold only in the thermodynamic limit. The saturation value of $S_{AB}$ is expected to approach $S_{AB,\text{Page}}$ upon the increase of the system size.

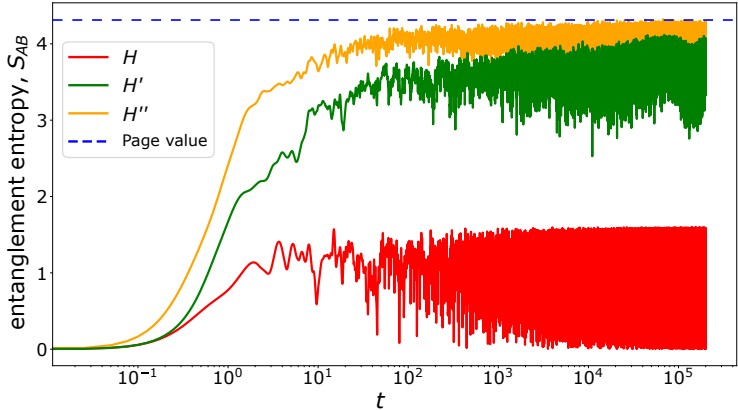

Figure 19: Growth of entanglement entropy $S_{AB}$ for an initial random state in $S^z$-basis in the $M = 6, M^x = M^y = (1, 1, \ldots, 1)$ symmetry sector of the $6 \times 6$ lattice with o.b.c., for different interaction ranges. $S_{AB}$ saturates to different values below the Page value for this symmetry sector, $S_{AB,\text{Page}} \approx 4.29$, at a rate that increases upon the increase of the interaction range. The system under $\hat{H}$ explores a 13-dimensional fragment of the Hilbert space and exhibits periodic behavior with perfect revivals (the first one is at $t \approx 3553$), while under $\hat{H}'$ and $\hat{H}''$ it explores the whole 720-dimensional symmetry sector. Despite the fact that the explored Hilbert space for $\hat{H}'$ and $\hat{H}''$ is the same, the saturation value of $S_{AB}$ is larger for $\hat{H}''$, presumably since the connectivity of the Hilbert space is enhanced (while it is not clear if the green line ever saturates to the same value as the yellow one, we confirm numerically that it does not do so at least until $t = 2.0 \cdot 10^5$).

from 1 to 13. We initialize the system in one of the $S^z$ basis states from the 13-dimensional fragment. Upon the increase of the interaction from $\hat{H}$ to $\hat{H}'$, all fragments merge into one 720-dimensional fragment. Upon the change of interaction from $\hat{H}'$ to $\hat{H}''$, the explored Hilbert space does not become bigger (all 720 states are already connected), but the improvement of the connectivity between the states leads to the increase of the saturation value of the entanglement entropy $S_{AB}$.

# 6 Discussion

**Generalization to other lattices and 3D:** We expect qualitatively similar physics in models on other lattices with ring-exchange interactions compatible with subsystem symmetries. In Fig. 20, on the three lattices we show the ring-exchange terms preserving subsystem symmetries, together with the 1D submanifolds on which these symmetries are defined.

On the honeycomb lattice, the minimal interaction comprise 4-site ring-exchanges on each hexagon, which upon increasing the interaction range can be elongated only in one direction, as shown on the right of Fig. 20(a). The generators of the corresponding subsystem symmetries are sums of $\hat{S}^z$ operators (as is always the case with ring-exchange interactions) along the zigzag lines shown on the left of Fig. 20(a). Note that if we were to consider the 6-site hexagon flip as the only term instead, the magnetization on each hexagon would be preserved, leading to an extensive number of local conserved quantities, which is much more restrictive than subsystem symmetries.

On the triangular lattice, the subsystem symmetries with support on the straight lines shown on the left of Fig. 20(b) allow for the hexagon flips, together with various longer-range

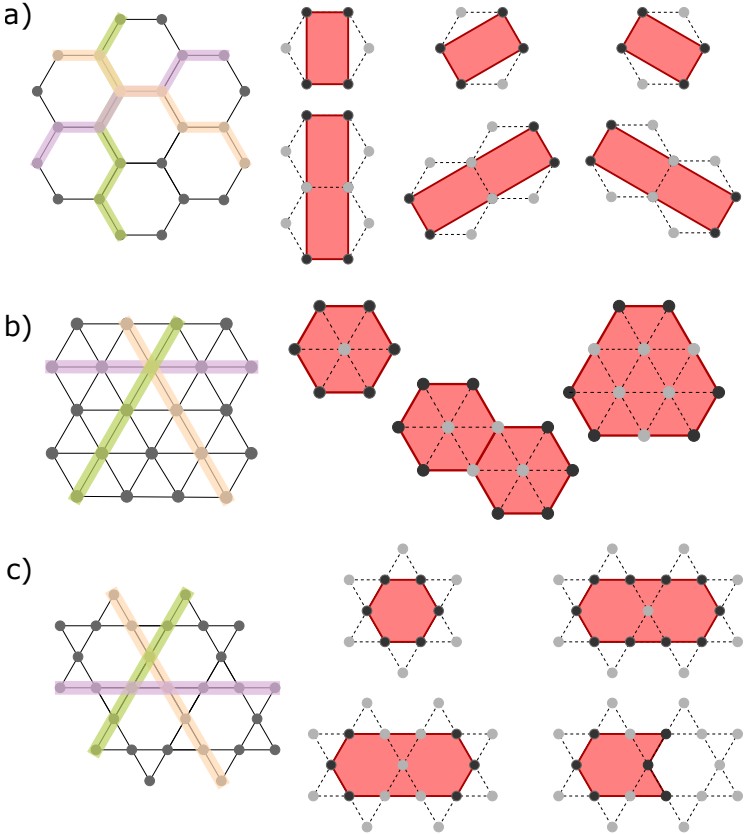

Figure 20: Ring-exchange terms depicted with red solid figures that preserve subsystem symmetries with support on 1D submanifolds depicted with yellow, purple and green lines, on (a) honeycomb, (b) triangular, (c) kagome lattices. Only spins colored in black take part in the ring-exchange, while light-grey spins do not. In addition to the most local interactions, several longer-range examples are shown.

ring-exchanges, as depicted on the right of Fig. 20(b). Analogously, one can do the same analysis for the kagome lattice [Fig. 20(c)] and various other 2D lattices.

Last but not least, one can straightforwardly make a generalization to 3 dimensions. Here subsystem symmetries can be defined not only on 1D, but also on 2D submanifolds. For instance, on the cubic lattice, two main possibilities arise. The first one concerns the same ring-exchanges as for the square lattice but on the $xy$, $xz$, and $yz$ planes of the cubic lattice. These interactions preserve magnetization on each plane, as well as the charge and dipole moment. A second possibility is to simultaneously flip spins at the vertices of an elementary cube (and the longer-range versions). This interaction preserves the magnetization not only in each plane, but also in each line parallel to $x, y, z$ directions. In addition, it conserves the quadrupole moment.

**Conclusions:** Our work explores the phenomenon of extensive Hilbert space fragmentation (shattering) in 2D. In addition to the conserved dipole moment, the class of models we consider have $\mathcal{O}(N)$ subsystem symmetries (with $N$ the linear size of the system). One might argue that $\mathcal{O}(1)$ global conserved quantities (charge and dipole moment conservation) are sufficient to observe Hilbert space fragmentation and that subsystem symmetries are therefore an unnecessary additional restriction on the dynamics. We show that the subsystem symmetries in conjunction with locality add an intricate feature to the shielding structures, that is their thickness depends only on the interaction range and not on the size of the active region which is being isolated. Therefore, it becomes possible to isolate infinite active regions with

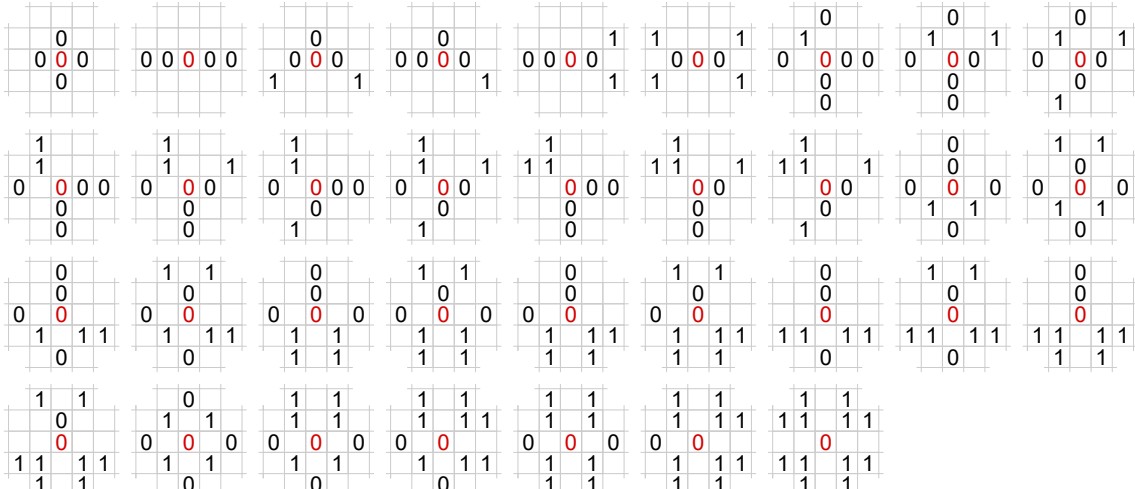

Figure 21: Local configurations of spins (in black) that are sufficient for the central spin (red) to not be immediately flippable under the action of the Hamiltonian $\hat{H}'$, i.e., rectangular interactions $1 \times 1, 1 \times 2, 2 \times 1$. All possible such configurations (330 in total) can be obtained from the ones presented in the figure by applying all possible combinations of $C_4$ and $C_2$ rotations, mirror reflections about the horizontal, vertical and two diagonal (at 45°) lines going through the central spin, as well as $\mathbb{Z}_2$ spin-flip.

finitely thick shields. This property might come in hand for applications in quantum memory, since it allows to protect an arbitrary region from decoherence using thin enough pre-designed shielding buffers.

In addition, as we have shown, subsystem symmetries alone do not produce an exponential number of inert states, as well as they are not enough to make the system integrable. This makes the kinetic constraints imposed by the ring-exchange interaction crucial for the extensive Hilbert space fragmentation. Despite this, there still exist large Hilbert space sectors. Thus, the XY-plaquette model exhibits versatile dynamical regimes, including abundant inert states and low-dimensional integrable sectors, as well as high-dimensional sectors with non-ergodic dynamics.

## Acknowledgements

The authors acknowledge the use of the computational facilities at the University of Zurich (UZH). A.K. is grateful to N. Astrakhantsev and C. Mudry for the discussions at the early stages of the project, to A. Davydov for the help with the computational cluster, and to M. Gavrilova for useful insights on integer sequences. A.T. acknowledges funding by the European Union's Horizon 2020 research and innovation program under the Marie Sklodowska Curie grant agreement No 701647. This work is supported in part by the DOE Grant No. DE-FG02-06ER46316 (A.K. and C.C.).

## A   Shielding structures for longer-range interactions

In Section 3, we discussed shields that can isolate active regions from each other, for $1 \times 1$ rectangular interactions. Analogous shields can be constructed for interactions of longer range.

For the Hamiltonian $\hat{H}'$ (i.e., rectangular interactions $1 \times 1, 1 \times 2, 2 \times 1$), we list all local

environments that are sufficient for the central spin to not be immediately flippable (Fig. 21). These environments can be constructed by adding additional spins to the environments from Fig. 10, which would ensure that none of the $2 \times 1$ and $1 \times 2$ plaquettes involving the central spin are immediately flippable. For $\hat{H}'$, there are in total 330 such local configurations. In Fig. 21 we list only the basic ones, while the rest can be obtained from them by applying symmetry operations: $C_4$ and $C_2$ rotations, mirror reflections about the horizontal, vertical or two 45°diagonal lines, and the flip of all spins.

Although now, using Fig. 21, we can construct a shield, i.e., a set of spins inert at all times, it is not enough to ensure the isolation of active regions. For example, imagine a simple shield that is a single straight line of 0's going through p.b.c. and separating the sample into two disconnected active regions. These two regions are not disentangled from each other, since the $1 \times 2$ interaction can still "penetrate" the shield, i.e., act on spins that are in different active regions. To make sure that the active regions are isolated, we need to make the shield two spins thick. In the simplest case it would be two adjacent lines of 0's.

Generically, we can conclude that in the XY-plaquette model active regions of arbitrary size can be isolated from each other by placing shields (sets of spins inert at all times) in between them that are as thick as the largest dimension of the interaction.

# B Determining the fragment size from time evolution

Here we show how one can deduce the dimensionality $d$ of a fragment $\mathcal{F}$ from the time evolution of a pure state $\psi_0 \in \mathcal{F}$. Under the Hamiltonian, $H$ the state evolves as $|\psi(t)\rangle = e^{-iHt} |\psi_0\rangle$. Let us project the resulting state $|\psi(t)\rangle$ to an arbitrary pure state $|\phi\rangle \in \mathcal{F}$ from the same fragment, $\langle \phi | \psi(t) \rangle = \langle \phi | e^{-iHt} | \psi_0 \rangle$. Taking the FT of the absolute value square of this projection,

$$\text{FT}\Big[|\langle \phi | \psi(t)\rangle|^2\Big](\omega) = \frac{1}{2\pi} \int_{-\infty}^{+\infty} \big|\langle \phi | e^{-iHt} | \psi_0 \rangle\big|^2 e^{-i\omega t} \, dt \; . \tag{17}$$

We can now diagonalize the Hamiltonian, $-iHt = PDP^{-1}$, where $P$ is independent of $t$ and $D$ is a diagonal matrix consisting of $-it\lambda_i$ with $\lambda_i$, $i = 1, \ldots, d$ the eigenvalues of $H$. The exponential $e^{-iHt}$ is then diagonalized by the same matrix $P$: $e^{-iHt} = Pe^D P^{-1}$. It is therefore clear that the expression under the integral in (17) will take the form

$$\big|\langle \phi | e^{-iHt} | \psi_0 \rangle\big|^2 e^{-i\omega t} = c_0 + \sum_{i \neq j} c_{ij} e^{-it(\lambda_i - \lambda_j + \omega)} \,, \tag{18}$$

and after the integration the peaks in the Fourier spectrum will be positioned at frequencies $\omega = 0, \lambda_1 - \lambda_2, \ldots, \lambda_i - \lambda_j, \ldots$ Assuming none of the energy differences are the same, we get $1 + \binom{d}{2}$ peaks in total (an additional overlap of peaks might happen if one of the eigenenergies is 0, while two other ones are negatives of each other). In order to avoid coinciding peaks, one could apply random magnetic field, which will shift every energy level in a non-generic way and resolve all $1 + \binom{d}{2}$ peaks in the Fourier spectrum.

However, another issue might arise. Depending on the states $|\psi_0\rangle, |\phi\rangle$, some of the coefficients $c_{ij}$ might be zero. Then, the protocol to determine $d$ should include several trials with different random states $|\psi_0\rangle$ and $|\phi\rangle$ and choosing the one with maximal number of peaks.

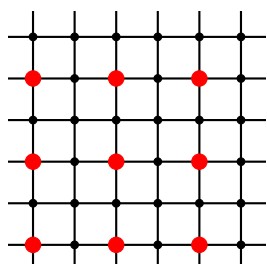

Figure 22: Red dots represent the set of sites on which the operator $\hat{O}$ from Eq. 19 is defined. Each $1 \times 1$ plaquette touches exactly one red site. Such a set exists only if $N_x, N_y$ are both even.

## C   Particle-hole symmetry

There exists an anticommuting operator for the Hamiltonian $\hat{H}$ (i.e., interaction range $1 \times 1$):

$$\hat{O} = \sum_{i \in \{\bullet\}} \hat{S}_i^z, \qquad \{\hat{O}, \hat{H}\} = 0, \tag{19}$$

where $\{\bullet\}$ is the set of sites depicted in Fig. 22. Each term in $\hat{H}$ contains exactly one operator belonging to this set. Note however that it is possible to construct such a set only if both $N_x$ and $N_y$ are even. As a result, the spectrum of the model with $1 \times 1$ interactions is symmetric with respect to $E = 0$ when $N_x, N_y$ are even. The operator $\hat{O}$ commutes with the subsystem symmetries, $[\hat{O}, \hat{U}_n^{x(y)}] = 0$, and therefore the energy spectrum of each symmetry sector is symmetric with respect to $E = 0$. It is also worth mentioning that although for any eigenstate $\psi$ with eigenenergy $E$ there exists another eigenstate $\tilde{\psi}$ with eigenenergy $-E$, the bipartite entanglement entropies of these two eigenstates need not be the same. Additionally, note that adding longer-range interactions destroys this particle-hole symmetry.

## D   From quantum link model to XY-plaquette model

In this Appendix we show how the XY-plaquette model can be obtained by lifting charge conservation in the QLM, which is a 2D lattice gauge theory [58, 59]. The fragmentation of the Hilbert space in the QLM has been shown in Ref. [44], where the authors study the localization-delocalization transition for certain initial conditions by associating the process with a percolation problem.

The model is defined on a square lattice with spin-1/2 d.o.f. positioned on the links, as shown in Fig. 23(a) (left panel). The $S^z$-basis states are denoted as arrows along the link. A plaquette is flippable if the four arrows around it are oriented clockwise or anticlockwise. The Hamiltonian for the QLM reads

$$\hat{H}_{\text{QLM}} = \lambda \sum_{\square} \left( U_{\square} + U_{\square}^\dagger \right)^2 - J \sum_{\square} \left( U_{\square} + U_{\square}^\dagger \right), \tag{20}$$

with $U_{\square} = \hat{S}_i^+ \hat{S}_j^- \hat{S}_k^+ \hat{S}_l^-$, where $i, j, k, l$ are sites around the plaquette. The first $\lambda$-term is just the potential energy which for $\lambda < 0$ favors configurations with many flippable plaquettes. Therefore, it does not play a role in the kinetic connectivity of basis states, unlike the second $J$-term which is the ring-exchange interaction, same as in (1). Thus, for our purposes, we consider QLM with $\lambda = 0$.

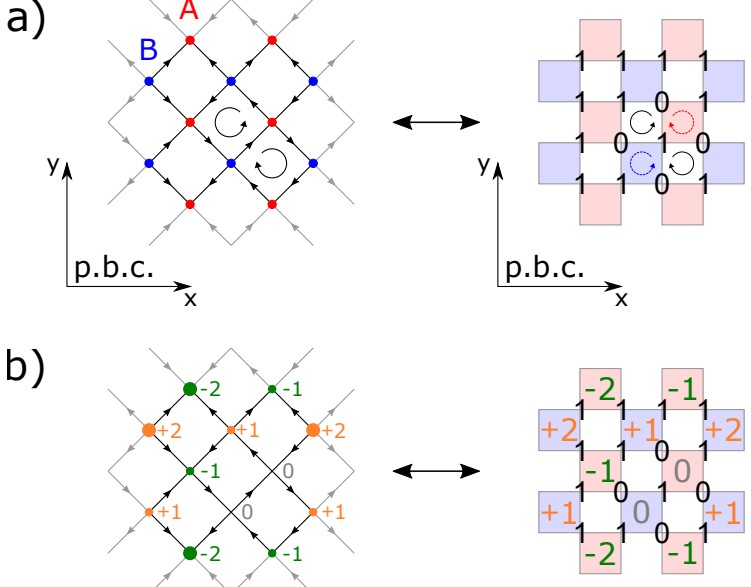

Figure 23: To see the connection between the QLM and the XY-plaquette model, it is convenient to represent QLM, that is formulated in terms of arrow d.o.f. on the links (left column), as a plaquette model with 0/1 d.o.f. on the sites of a square lattice rotated by 45° with respect to the original lattice, which is a natural framework for the XY-plaquette model (right column). (a) The vertices of the QLM are bipartitioned into A (red) and B (blue) sublattices. In the language of the XY-plaquette model, the number of plaquettes is doubled, with white plaquettes corresponding to the original plaquettes of the QLM, and blue/red plaquettes corresponding to the vertices of the QLM. Unlike in the QLM, in the XY-plaquette model we can additionally flip blue and red plaquettes. (b) Charges are defined on the vertices of the QLM in (21), i.e., on blue/red plaquettes in the XY-plaquette model language. They are conserved in the QLM, while this conservation is lifted when we allow blue/red plaquettes to flip.

To establish the correspondence between the arrow notation and $\left|s^z = -\frac{1}{2}\right\rangle \equiv |0\rangle$, $\left|s^z = +\frac{1}{2}\right\rangle \equiv |1\rangle$ notation, we have to divide the lattice into two sublattices, A and B, as shown in Fig. 23(a) (left panel). Then on sublattice A we define "arrow in vertex" $\equiv |1\rangle$, "arrow out of vertex" $\equiv |0\rangle$; and vice versa on sublattice B: "arrow in vertex" $\equiv |0\rangle$, "arrow out of vertex" $\equiv |1\rangle$. It is then straightforward to slightly redraw the lattice, so that it takes a more familiar form of a (0,1)-matrix. On the right panel of Fig. 23(a), white plaquettes correspond to the original plaquettes on the left panel, while red and blue plaquettes correspond to red and blue vertices on the left panel. If we formulate QLM as on the right panel of Fig. 23(a), the terms in the Hamiltonian (20) are defined only on white plaquettes, and therefore the XY-plaquette model is the QLM (with $\lambda = 0$) with more dynamics allowed, since it adds the possibility for the red and blue plaquettes to be flipped. We will further show how this additional dynamics introduces fracton-like restricted mobility for the charges that were conserved in the QLM. Before that, two important notes are in place: first, for the XY-plaquette model to have the conventional p.b.c. of a square lattice, the p.b.c. in the QLM have to be imposed not along the links, but rather along the diagonals of the plaquettes, as shown by the $x$ and $y$ directions in Fig. 23(a); second, such a mapping can only result in a $2n \times 2n$ lattice, since otherwise it is not possible to bipartite the lattice into A and B sublattices.

Now, let us identify conserved quantities in the QLM. We define charge at site $i$ as

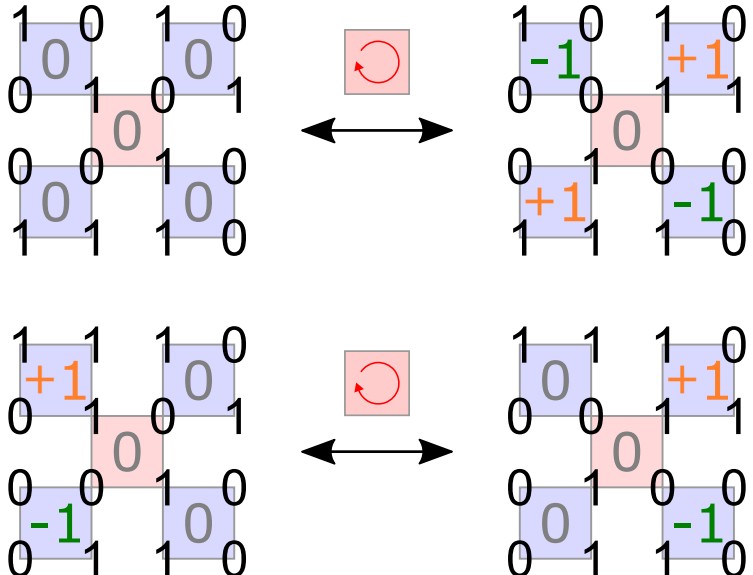

Figure 24: Fracton-like dynamics of the charges on the blue sublattice upon the flip of the plaquette on the red sublattice: (top panel) new charges can be created only in quadrupoles, which are free to move in the plane, (bottom panel) a dipole of charges can move only in the direction perpendicular to its dipole moment. A single charge remains immobile under the dynamics of the XY-plaquette model.

$q_i = \frac{1}{2}(\text{arrows out} - \text{arrows in})$, which correspond to charge operators

$$\hat{q}_i = (-1)^\eta \sum_{+} \hat{S}_i^z, \tag{21}$$

where the summation goes over four links adjacent to the vertex $i$, $\eta = 1$ for sublattice A and $\eta = 0$ for sublattice B (see Fig. 23(b)). These operators commute with the Hamiltonian at each site, $[\hat{H}_{\text{QLM}}, \hat{q}_i] = 0$, and therefore the charge at each vertex in the QLM is conserved. Another set of conserved quantities is the fluxes, defined on the vertical (horizontal) lines (along $y$ ($x$)-direction) going through the middle of the links, $\Lambda_n^{x(y)}, n = 1, \ldots, N_x(N_y)$. We define it as

$$\Phi_n^x = \frac{1}{2}(\text{arrows to the right} - \text{arrows to the left}),$$
$$\Phi_n^y = \frac{1}{2}(\text{arrows up} - \text{arrows down}), \tag{22}$$

and the corresponding symmetry operators:

$$\hat{\Phi}_n^x = (-1)^n \sum_{i \in \Lambda_n^x} \hat{S}_i^z,$$
$$\hat{\Phi}_n^y = (-1)^n \sum_{i \in \Lambda_n^y} \hat{S}_i^z, \tag{23}$$

where $\Lambda_n^x$ with an even $n$ is such that the sublattice A lies to the right and the sublattice B lies to the left of the line $\Lambda_n^x$ (analogously for $\Lambda_n^y$: $n$ is even when the sublattice A lies above the line $\Lambda_n^y$ and the sublattice B lies below the line $\Lambda_n^y$). The fluxes are connected to the charges

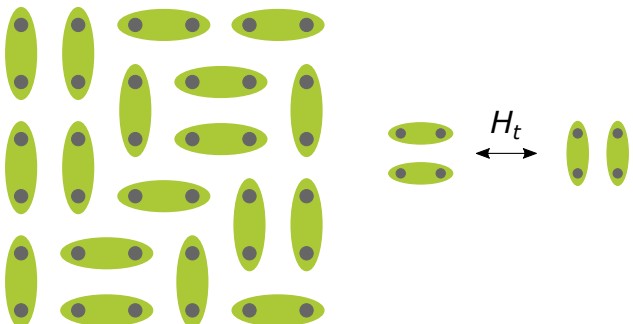

Figure 25: Fully packed QDM. Each site is dimerized with one of its nearest neighbors. Two parallel nearest neighbor dimers can flip, as shown on the right.

through the Gauss's law:

$$
\begin{aligned}
\hat{\Phi}^x_{n+1} - \hat{\Phi}^x_n &= \sum_{\substack{j \in \text{between} \\ \Lambda^x_n \text{ and } \Lambda^x_{n+1}}} \hat{q}_j\,, \\
\hat{\Phi}^y_{n+1} - \hat{\Phi}^y_n &= \sum_{\substack{j \in \text{between} \\ \Lambda^y_n \text{ and } \Lambda^y_{n+1}}} \hat{q}_j\,.
\end{aligned}
\tag{24}
$$

When we transition from the QLM to the XY-plaquette model by allowing red and blue plaquettes to flip, the fluxes $\hat{\Phi}^x_n, \hat{\Phi}^y_n$ remain conserved (in fact, they become the subsystem symmetries defined in (2)), while we lift the conservation of charges $\hat{q}_i$. However, although the charges become dynamical quantities, they still possess local charge and dipole moment conservation on each sublattice. We illustrate this in Fig. 24, where we observe what happens to the charges on the blue sublattice when the red plaquette is flipped. One can see that new charges can be created only in quadrupoles, while a dipole can freely move along the direction perpendicular to its dipole moment. In turn, a single charge is immobile.

# E Equivalence between the quantum dimer model and the quantum link model

In this Appendix we show how the fully packed QDM is equivalent to the QLM in a particular vertex-charge sector. The QDM is defined on a square lattice where every site is dimerized with one of its nearest neighbors, as shown in Fig. 25. The corresponding Hamiltonian is

$$
\begin{aligned}
\hat{H}_{\text{QDM}} &= \hat{H}_V + \hat{H}_t\,, \\
\hat{H}_V &= V \sum_{\square} \left( |\text{◖◗}\rangle\langle\text{◖◗}| + |\text{⬭}\rangle\langle\text{⬭}| \right), \\
\hat{H}_t &= -t \sum_{\square} \left( |\text{◖◗}\rangle\langle\text{⬭}| + |\text{⬭}\rangle\langle\text{◖◗}| \right).
\end{aligned}
\tag{25}
$$

The equivalence can be easily seen if we divide all horizontal and vertical links in QDM into two sublattices, A and B. Then, a dimer on sublattice A we identify with an arrow to the right or down, while the absence of a dimer we identify with an arrow to the left or up. We do the opposite on the sublattice B. In this manner, we have rewritten the QDM in the language of QLM. Since each site in the fully packed QDM belongs to exactly one dimer, each vertex in the corresponding QLM will have staggered $\pm 1$ vertex charges. $\hat{H}_V$ and $\hat{H}_t$ are exactly equal

to the $\lambda$- and $J$-terms in the QLM, respectively. Therefore, the fully packed QDM is a special instance of the QLM, taken in a particular charge sector.

In principle, one can consider defects in the QDM: monomers (sites that are not paired up in dimers) and higher-order polymers, when more than two spins are entangled (however, we do not allow long-range entangled dimers or polymers). With such defects, the QDM becomes exactly equivalent to the QLM, allowing a configuration in any charge sector.

## F  Bound on the number of inert states with infinite range interactions

For completeness, here we explicitly show how to obtain the bound from Eq. 12.

$$
\begin{aligned}
\sum_{j=1}^{N}(-1)^{N+j}j!\begin{Bmatrix}N\\j\end{Bmatrix}(j+1)^N &< \sum_{j=0}^{N}j!\begin{Bmatrix}N\\j\end{Bmatrix}(j+1)^N \leq \sum_{j=0}^{N}\frac{1}{2}\frac{N!}{(N-j)!}j^{N-j}(j+1)^N \\
&< \sum_{j=0}^{N}\frac{1}{2}N^j j^{N-j}(j+1)^N = \sum_{j=0}^{N}\frac{1}{2}\left(\frac{N}{j}\right)^j (j(j+1))^N < \sum_{j=0}^{N}\frac{1}{2}\frac{N^j}{j!}(j(j+1))^N \\
&< \frac{1}{2}\left(N(N+1)\right)^N \sum_{j=0}^{N}\frac{N^j}{j!} \xrightarrow{N\to\infty} \frac{1}{2}N^{2N}e^N,
\end{aligned}
\tag{26}
$$

where in the first line we have used the upper bound on the Stirling number of the second kind from Ref. [72]:

$$
\begin{Bmatrix}N\\j\end{Bmatrix} \leq \frac{1}{2}\binom{N}{j}j^{N-j}.
\tag{27}
$$

## G  Locality and "shape" of interactions

In order to study the interplay of locality and the "shape" of subsystem symmetry preserving interactions, consider increasing the interaction range to infinity. Does one have to include all subsystem symmetry preserving interactions in order to completely remove the Hilbert space fragmentation? The answer is no. There is a minimal set of interactions that are sufficient to dynamically connect all the states within each of the symmetry sectors. On the square lattice, such set consists of "rectangular" interactions (8). This is apparent, since, as we have stated before, the construction of all states in a particular symmetry sector is performed by recursively flipping all flippable rectangles. This procedure necessarily will produce all states with prescribed column and row sums, due to the theorem from Ref. [63]. The existence of such a minimal set implies that if in a physical system, for whatever reason, some of the interactions from this set are forbidden (or exponentially suppressed in comparison to interactions of other "shapes" of the same range), the Hilbert space fragmentation would persist even upon increasing the interaction range to infinity.

Let us demonstrate this on an example. If one interprets the ring-exchange term as a hopping of hard-core bosons, it seems plausible that in the presence of a strong on-site potential, the correlated hopping should only occur between the neighboring sites, and therefore only the terms like $\hat{H}_{1\times 1}$ and

$$
\begin{pmatrix} 0 & 1 & 0 & 1 \\ 1 & \bullet & \bullet & 0 \\ 0 & \bullet & \bullet & 1 \\ 1 & 0 & 1 & 0 \end{pmatrix} \longleftrightarrow \begin{pmatrix} 1 & 0 & 1 & 0 \\ 0 & \bullet & \bullet & 1 \\ 1 & \bullet & \bullet & 0 \\ 0 & 1 & 0 & 1 \end{pmatrix}
\tag{28}
$$



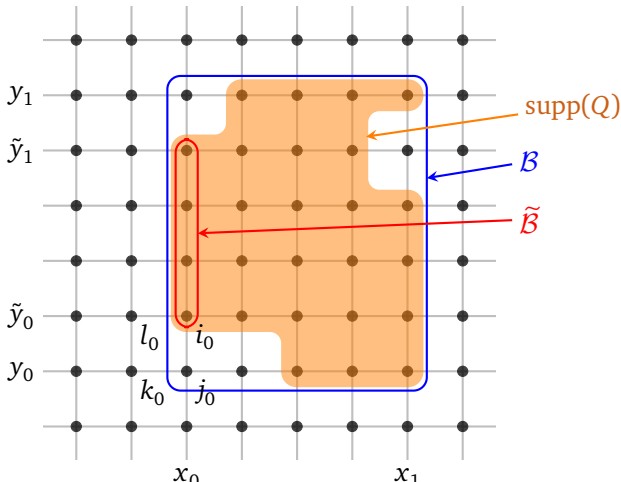

Figure 26: Support of a local operator $Q$, supp($Q$) (in orange), is fully located inside a rectangle $\mathcal{B} \equiv [x_0, x_1] \times [y_0, y_1]$ (in blue). The intersection of supp($Q$) and the leftmost column of $\mathcal{B}$ is a set $\tilde{\mathcal{B}} \equiv \{x_0\} \times [\tilde{y}_0, \tilde{y}_1]$ (in red). Since $Q$ is local, we can always identify the lowermost site $i_0$ of the rightmost column of the support of $Q$.

are not exponentially suppressed, i.e., the terms where the ring-exchange acts on a loop with no "gaps" in between neighboring flipped spins. In such a setting, even upon an increase of the interaction range to infinity, the Hilbert space fragmentation will not disappear, since generic rectangular interactions (8) are not allowed. In particular, for instance, states (6) will not become connected to each other.

While the generic mechanisms leading to the fragmentation of the Hilbert space in quantum models are still not clear, our observation points to the fact that in real physical systems the locality of interactions in itself may not be a necessary ingredient for it. It is also important to take into account the geometry of interactions allowed in a concrete system.

## H   Absence of local conserved quantities

One might ask whether the studied XY-plaquette model has any emergent local conserved quantities. In case it does, they might be responsible for the observed Hilbert space fragmentation. In this Appendix, we prove that no local conserved operator can exist in the XY-plaquette model.

The proof goes as follows. Let us assume that there exists a local conserved operator $Q$, i.e., $[H, Q] = 0$. By "local" we mean that the support of the operator, supp($Q$), is bounded. Thus, on the considered square lattice, supp($Q$) is fully located inside some rectangle $\mathcal{B} \equiv [x_0, x_1] \times [y_0, y_1]$, i.e., supp($Q$) $\subseteq \mathcal{B}$. Therefore, supp($Q$) consists of at most $P = (x_1 - x_0 + 1)(y_1 - y_0 + 1)$ sites.

Since the local degrees of freedom obey Pauli algebra, the operator $Q$ generically can be written as

$$Q = \sum_{\mu} c_{\mu} \bigotimes_{i=1}^{P} \sigma_i^{\mu_i}, \tag{29}$$

where $\mu \equiv \{\mu_1, \mu_2, \ldots, \mu_P\}$ is a string composed of $\mu_i \in \{0, 1, 2, 3\}$, such that $\sigma_i^{\mu_i}$ is one of the four Pauli matrices (including identity) at site $i = 1, \ldots, P$. The summation goes over all $4^P$ possible strings and $c_{\mu}$ are arbitrary complex coefficients.

Consider the intersection of the support of $Q$ and sites at $x = x_0$, i.e., sites in the leftmost column of $\mathcal{B}$. Since supp($Q$) is bounded, this intersection, supp($Q$)$\cap\{x = x_0\}$, is also bounded and lies inside region $\tilde{\mathcal{B}} \equiv \{x_0\} \times [\tilde{y}_0, \tilde{y}_1]$, where $\tilde{y}_0 \geq y_0$, $\tilde{y}_1 \leq y_1$. Thus, we can identify site $i_0 \equiv (x_0, \tilde{y}_0)$. In short, for any local operator $Q$ on a square lattice, we can always find the lowermost site of the leftmost column belonging to the support of $Q$ (see Fig. 26).

This, in turn, means that there exist terms in the Hamiltonian that have overlap with supp($Q$) on one site only. In particular, terms $\sigma_{k_0}^+ \sigma_{l_0}^- \sigma_{i_0}^+ \sigma_{j_0}^-$ and $\sigma_{k_0}^- \sigma_{l_0}^+ \sigma_{i_0}^- \sigma_{j_0}^+$ overlap supp($Q$) only on site $i_0$, as shown in Fig. 26. This fact, together with $[H, Q] = 0$ and the form of the operator $Q$, (29), leads to the conclusion that for any non-zero term in (29), $\mu_{i_0} = 0$. I.e., $Q$ can only contain an identity acting on site $i_0$. However, this contradicts the fact that $i_0$ belongs to the support of $Q$. Therefore, our initial assumption about the existence of a local conserved operator $Q$ was wrong.

Note that the addition of arbitrary $\sigma^z$ terms to the Hamiltonian does not alter the conclusion.

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
