# Peer review of "Hilbert space fragmentation in a 2D quantum spin system with subsystem symmetries"

_SciPost Physics, doi:SciPost Phys. 13, 098 (2022)_

## Round 2 · Referee Report · Rahul Nandkishore (Referee 1) · 2022-2-7

Strengths

Very clear and understandable paper, extending the phenomenon to a broader class of models. There is a clear `new result' viz that for subsystem symmetries, finite range shields can disconnect arbitrarily large active regions.

Weaknesses

The authors do not always clearly distinguish what phenomena have only been checked in finite size numerics, and what have been demonstrated to survive even in the thermodynamic limit, leaving this to the reader to infer. It would be good to clarify this in the text. (I think though that their main `new' result re: finite size shields does survive the thermodynamic limit).

Report

The paper studies Hilbert space fragmentation (shattering) in a family of spin models with ring exchange terms, which have subsystem symmetry. The models are studied for tunable interaction range, where the interactions commute with the subsystem symmetry operators. Fragmentation into an exponentially large number of subsectors is demonstrated, for finite interaction range, but not for infinite interaction range. A key result is that the shielding regions need only be as large as the interaction range, unlike the case of dipole conserving models where the shielding regions need to be as large as the region being shielded.

This is a good, solid, useful paper, and I recommend publication. There are however some points that I would encourage the authors to address - see below.

  1. Above Eq.6, should the symmetry sector label be Mx=(1,0,1,0) instead of what is currently written?
  2. Below Eq.9 it is noted that the boundaries of the figures can be self intersecting. one might worry that for self intersecting figures, it would be ambiguous how to alternate raising and lowering operators. An example here might be useful.
  3. Did the authors think about whether the fragmentation here is strong or weak? i.e., does a randomly chosen product state appear to thermalize, or not? Given the existence of finite sized shielding regions, which can disconnect the system into disentangled parts (Fig.12), one might expect that there is a possibility of strong fragmentation here, which would be a striking result in two space dimensions. This would be worth checking, and, if it could be convincingly argued to survive the thermodynamic limit, would be worth advertising more prominently in abstract and introduction as well.
  4. Did the authors think about the hydrodynamics of charge in their models? It should be subdiffusive (see e.g. Ref.69 and also PRR 2, 033124 (2020). Presumably it is hard to extract hydrodynamics for the Hamiltonian models, because of system size constraints, but it should be possible for the natural generalization to automaton dynamics, using the Monte Carlo technique from Ref. 69. (This might be beyond the scope of the present work, but a worthwhile project to consider for the future).
  • validity: high
  • significance: high
  • originality: good
  • clarity: top
  • formatting: excellent
  • grammar: excellent

Author:  Alexey Khudorozhkov  on 2022-04-02  [id 2349]

(in reply to Report 1 by Rahul Nandkishore on 2022-02-07)

Dear Prof. Nandkishore,

Thank you for the insightful comments. Let us address the four points you raised.
  1. We count the quantum numbers from left to right in \(M^x\) and from bottom to top in \(M^y\). We have added the corresponding clarification in the text. The quantum numbers above Eq. (6) are therefore correct.

  2. Indeed, the assignment of spin-raising and spin-lowering operators for self-intersecting figures might be confusing. The main idea for constructing a figure is the following: start from a site -> go to any other site in the same row -> go to any other site in the same column -> go to any other site in the same row -> go to any other site in the same column -> ... and so on. This can lead both to figures with and without self-intersection. We have added a third example in Eq. (9), which has self-intersection. Hopefully, it clarifies the issue.

  3. The question of strong vs weak fragmentation is indeed very interesting. Note that the shields in our model are necessarily non-local: they have to go through the periodic boundary or terminate at the open boundary. Therefore, in the thermodynamic limit, the probability of having a shield in a random product state tends to zero. We have added the corresponding statement to the paper. Shields are a sufficient but not necessary condition for the Hilbert space fragmentation. Therefore, it is difficult to argue whether the XY-plaquette model exhibits weak or strong fragmentation, since the absence of shields in a typical product state does not imply that this state thermalizes. We believe that there is little hope that the XY-plaquette model has strong fragmentation. However, we cannot rigorously prove this, and therefore we will avoid making any claims in the paper. In our opinion, it is an interesting venue of research, to try to construct a 2D model with only global and/or subsystem symmetries that would support local shields. In this case, the probability of a random product state having a shield in the thermodynamic limit is non-zero, and strong fragmentation is guaranteed.

  4. Since all the numerical results in the paper are based on exact diagonalization, the reachable system sizes are small, and probing hydrodynamics does not seem conceivable. Indeed, as you point out, the interaction term from the Hamiltonian in Eq.(1) satisfies the automaton constraint, and therefore the model can be studied using automaton dynamics. This is beyond the scope of the current paper, but we would be interested in discussing this topic with you in more detail.

Thank you!

---

## Round 2 · Referee Report · Anonymous (Referee 2) · 2022-2-21

Strengths

Very detailed study exploring various aspects of the system studied

Weaknesses

It remains unclear whether fragmentation or symmetries are responsible for ergodicity breaking

Report

I think that the paper is interesting and makes a contribution to a quickly growing topic in non-equilibrium quantum physics, namely fragmentation.

My main issue with the paper is that is unclear whether the observed lack of ergodicity is due to "true" fragmentation or the presence of additional local algebraic structures. If the former, then the results are complete. If the latter, then the paper needs to be rewritten because lack of ergocitity is not due to real fragmentation.

As explained in this paper: https://arxiv.org/abs/2108.10324 actual fragmentation is not due to local conservation laws, but rather more complicated algebraic structures. In fact as discussed in this paper: https://arxiv.org/abs/2108.13411 subsystem symmetries (or more exotic local algebraic structures) can induce fragmentation of the Hilbert space, but this is not "true" fragmentation in the sense of a novel Hilbert space structure beyond conservation laws. (The second paper calls this case local Hilbert space fragmentation).

The authors' system clearly has subsystem symmetries, does it have more local symmetries that may be responsible for the fragmentation? It remains unclear whether the system has this from the text.

Related to this, the authors state now: "...the shattering is not simply a consequence of the sub-extensive number of conserved quantities due to subsystem symmetries" (p. 5).

Why do subsystem symmetries imply a sub-extensive number of conserved quantities? This is very much unclear to me. What measure and what space of conservation laws are used? A product of two local subsystem symmetries is another (orthogonal wrt to the Kubo-Mori inner product) symmetry. I can make a super extensive number of conserved quantities by repeatedly taking products of local subsystem symmetries that are physically finitely separated.

Requested changes

  1. Check (e.g. numerically) for presence of additional local algebraic structures. This can be done numerically for a local enough operator. Or if this was already done, discuss more prominently and in more detail.

  2. Explain measure of subsystem symmetries as per the above.

  • validity: high
  • significance: good
  • originality: ok
  • clarity: high
  • formatting: perfect
  • grammar: excellent

Author:  Alexey Khudorozhkov  on 2022-03-29  [id 2332]

(in reply to Report 2 on 2022-02-21)
Category:
answer to question
correction

Here are our responses to both issues raised by the referee.

Issue 1.

We believe that the Hilbert space fragmentation that we observe in the XY-plaquette model is indeed a "true" fragmentation. We can show this in the following way. Table I in paper https://arxiv.org/abs/2108.10324 presents a classification of systems based on the scaling of the commutant algebra ( \(\mathcal{C}\)) dimension. As seen from the table, for 2D systems, the true fragmentation occurs if \(\log(\dim{\mathcal{C}})\) scales as \(N^2\) (where \(N\) is the linear size of the system), while a mere presence of continuous subsystem symmetries gives \(\log(\dim{\mathcal{C}}) \sim N \log{N}\) . We can argue that the XY-plaquette model exhibits the \(N^2\) scaling behavior.

First, note that, as we have stated in our paper, the XY-plaquette model generically accepts arbitrary \(S^z\) terms, since such terms do not violate any of the subsystem symmetries (as well as the charge and dipole moment conservation). Then, following the terminology of the same paper (https://arxiv.org/abs/2108.10324), the bond algebra of the XY-plaquette model, \(\mathcal{A}\), is generated by the operators of the form \(S^+S^-S^+S^-\) (+h.c.) from Eq.(1) of our paper and operators \(S_j^z\) on every site \(j\) (as well as the identity operator). As explained in Appendix A of https://arxiv.org/abs/2108.10324, this ensures that the commutant algebra \(\mathcal{C}\) consists only of operators diagonal in the product state basis (the authors call this scenario "classical fragmentation"). Therefore, the projectors onto Krylov subspaces (which we call "fragments" in our paper) form an orthogonal basis for \(\mathcal{C}\). Therefore, calculating \(\dim{\mathcal{C}}\) is equivalent to calculating the number of fragments, i.e., the number of disconnected Hamiltonian blocks in a product state basis (in our case, \(S^z\) basis).

In our paper, we show that the log of the number of 1-dimensional Krylov subspaces (inert states) scales as \(N^2\). This can be seen in Fig.4 (red line). Each inert state is associated with the corresponding projector onto it. As a result, this proves that \(\log(\dim{\mathcal{C}})\) scales at least as \(N^2\). In addition, we have changed the right panel of Fig. 2 (see attachment), where we now show that the number of symmetry sectors scales as \(N\log{N}\), which coincides with the result for subsystem symmetries from Table I in paper https://arxiv.org/abs/2108.10324. To sum up, in Fig. 2 we show that subsystem symmetries alone do not produce the \(N^2\) scaling required for "true" fragmentation, while in Fig. 4 we show that kinetic constraints of the Hamiltonian do produce the \(N^2\) scaling behavior. Therefore, the model exhibits "true" fragmentation, which is due to kinetic constraints and not due to subsystem symmetries.

This reasoning is in fact almost identical to the section VI of https://arxiv.org/abs/2108.10324, where the analysis is performed for a 1D spin-1 dipole-conserving model. In addition, we observe that for infinite-range interactions the previously observed blocks (red blocks in Fig.3) disappear, leaving only the blocks corresponding to different quantum numbers of subsystem symmetries (solid blue blocks in Fig.3), which further shows that the extensive \(N^2\) fragmentation of the Hilbert space is not due to subsystem symmetries.

We think that checking numerically for the absence of additional local conserved quantities is not required for showing that the model has "true" fragmentation.

Issue 2.

When we say "The number of independent symmetries is \(n\).", we mean that the minimal number of generators of the (multiplicative) group of these symmetry operators is \(n\). This is a natural definition, since it answers the question "How many independent quantum numbers do we have to provide in order to unambiguously fix the symmetry sector?". Of course, as the referee pointed out, if there are two symmetry operators, \(O_1\) and \(O_2\), then the operator \(O_1 O_2\) also commutes with the Hamiltonian and is therefore a symmetry operator. However, if a given state \(\left| \psi \right>\) is an eigenstate of \(O_1\) with an eigenvalue \(\lambda_1\) and an eigenstate of \(O_2\) with an eigenvalue \(\lambda_2\), then \(\left| \psi \right>\) is automatically an eigenstate of \(O_1O_2\) with an eigenvalue \(\lambda_1\lambda_2\), which is uniquely determined given \(\lambda_1\) and \(\lambda_2\).

In our case, as we say in the paper, on a square lattice of size \(N_x \times N_y\), the number of independent subsystem symmetries is \(N_x + N_y - 1\) (\(N_x\) column-sum operators, \(N_y\) row-sum operators, and -1 originating from the fact that the product of all column-sum operators is equal to the product of all row-sum operators). So, one has to fix \(N_x + N_y - 1\) quantum number to unambiguously fix the symmetry sector. On the other hand, the number of degrees of freedom is \(N_x N_y\). Therefore, we say that the number of subsystem symmetries is "sub-extensive", since it scales linearly with \(N\), rather than quadratically.

Attachment:

num_symm_sectors_new.pdf

---

## Round 3 · Referee Report · Rahul Nandkishore (Referee 1) · 2022-4-6

Report

I think the authors have adequately addressed the reports and the manuscript can now be published.

---

## Round 3 · Referee Report · Anonymous (Referee 2) · 2022-4-16

Report

The authors have made considerable efforts to check the conditions of https://arxiv.org/abs/2108.10324 for fragmentation based on commutation algebras.

However, as mentioned in my first report, as discussed in https://arxiv.org/abs/2108.13411 other local algebraic structures beyond subsystem symmetries (e.g. [H,A]=\omega A with A acting on a subsystem) can further induce fragmentation. Such a case would likewise not be "true" fragmentation and it is not clear to me from the authors expanded study whether there are such algebras in their system.

Requested changes

Check the conditions discussed above to provide further evidence that the system has true fragmentation.

  • validity: -
  • significance: -
  • originality: -
  • clarity: -
  • formatting: -
  • grammar: -

Author:  Alexey Khudorozhkov  on 2022-05-12  [id 2458]

(in reply to Report 2 on 2022-04-16)
Category:
answer to question
validation or rederivation

We have added Appendix H, where we prove analytically that the XY-plaquette model cannot exhibit local conserved quantities. Please see the resubmition.

---

## Round 3 · Author Response

Added minor clarifications in several sections of the paper. Changed one figure.

---

## Round 3 · List of Changes

1. Changed the left panel of Fig. 2. We changed the horizontal axis to Nlog(N). We show that the number of symmetry sectors is proportional to Nlog(N). This scaling behavior is expected for continuous subsystem symmetries, and therefore we prove that the subsystem symmetries alone are not responsible for Hilbert space fragmentation in the considered model. We added corresponding comments on this in Section II.
  2. At the beginning of Section III, we added a clarification on the notation for subsystem symmetry quantum numbers.
  3. In "Locality and fragmentation" subsection of Section III, in Eq. (9), we added another example of a possible interaction term that preserves subsystem symmetries. The "shape" of this interaction has intersecting edges. This example is supposed to clarify our comment on self-intersecting boundaries of the interaction "figures" below Eq. (9).
  4. In Section IV, we added an important statement, that non-locality of the shields implies that in the thermodynamic limit the probability of having a shield tends to zero.

---

## Round 4 · Referee Report · Anonymous (Referee 2) · 2022-5-13

Report

The authors have now clearly demonstrated their system has true fragmentation and I judge the main conclusion of the paper to be correct.

As I wrote in my first report the paper deals with an extensive study of a topical subject and the model studied is a nice addition to the quickly growing zoo of models with Hilbert space fragmentation. In this case the interplay between subsystem symmetries and fragmentation leads to a "shielding" effect, which is in itself new.

However, despite this I cannot find sufficient evidence of any of the SciPost Physics acceptance criteria being fulfilled by this work:

  1. Detail a groundbreaking theoretical/experimental/computational discovery;

  2. Present a breakthrough on a previously-identified and long-standing research stumbling block;

  3. Open a new pathway in an existing or a new research direction, with clear potential for multipronged follow-up work;

  4. Provide a novel and synergetic link between different research areas.

One could possibly argue for 3, but the model studied seems quite specific and, since the authors do not discuss any experimental implementation that could realize the model, I do not see "clear potential for multipronged follow-up work".

Based on this conclusion I recommend publication in SciPost Physics Core.

---

## Round 4 · Author Response

We address the concern raised by one of the referees about the existence of emergent local conserved quantities in the considered model. We have added Appendix H, where we present an analytical proof that the XY-plaquette model does not have any local conserved operators.

---

## Round 4 · List of Changes

• Added Appendix H, where we prove that local conserved quantities do not exist in the XY-plaquette model.

---

## Editorial Decision

published